# Cognitive Insights and Stable Coalition Matching for Fostering LLM-based Multi-Agent Cooperation

## Abstract

Cognitive abilities, such as Theory of Mind (ToM), play a vital role in facilitating cooperation in human social interactions. However, Large Language Model (LLM) agents with higher ToM abilities do not necessarily exhibit better cooperative trends compared to those with lower ToM abilities, highlighting the complexity of translating human cognitive processes to artificial agents. To address this challenge, we propose a novel matching coalition mechanism that leverages the strengths of agents with different ToM levels by explicitly considering belief alignment and specialized abilities when forming coalitions. Our proposed stable coalition formation algorithm seeks to find the team that maximizes the potential for cooperative trends and ensures long-term viability. By incorporating cognitive insights into the design of multi-agent systems, our work demonstrates the potential of leveraging ToM to create more sophisticated and human-like coordination strategies that foster cooperation and improve overall system performance.

## 1 Introduction

Cooperation is a fundamental aspect of multi-agent systems, enabling agents to work together effectively to achieve common goals and solve complex problems (Shenoy, 1979). In recent years, the rapid advancement of large language models (LLMs) has opened up new opportunities for building intelligent multi-agent systems. LLMs have demonstrated remarkable capabilities in natural language understanding, generation, and reasoning, such as GPT-family (Eloundou et al., 2023). By leveraging these powerful LLMs, multi-agent systems can enable agents to communicate and collaborate using natural language, resulting in more flexible and human-like interactions in cooperation tasks.

To foster cooperation among LLM-based agents, most recent research focused on communication and knowledge sharing among agents (Xu et al., 2023; Lan et al., 2023; Hua et al., 2023; Wu et al., 2023b; Nascimento et al., 2023; Fu et al., 2023). These studies demonstrate the potential of leveraging natural language capabilities to facilitate information exchange and coordination in multi-agent systems. Besides language understanding, some research has explored effective cooperation through agent cognitive abilities, such as reasoning and reflection, to coordinate actions and make decisions (Qi & Vul, 2020; Li et al., 2023b; Liang et al., 2023; Lin et al., 2024). These studies highlight the importance of investigating *how we can foster cooperation in LLM-based multi-agent systems from a cognitive perspective*.

One of the fundamental cognitive abilities is the Theory of Mind (ToM). By utilizing ToM in strategic interactions, an agent can mentally simulate others' thoughts and potential actions. Furthermore, ToM can involve multiple levels of recursive belief attribution, known as higher-level ToM, where players consider not only their opponent's beliefs but also their beliefs about the other player's beliefs, and so on (Premack & Woodruff, 1978). In other words, ToM allows one to see things from others' perspective. In human social interactions, ToM plays a crucial role in facilitating cooperation by enabling individuals to understand and predict the behavior of others (Yoshida et al., 2008). For example, in a chess game, a player with higher-level ToM might think, "I believe my opponent thinks that I will move my knight, so I will move my bishop instead, because I believe they will not expect that move." Normally, in a specific scenario, agents with higher level ToM can better understand and predict the actions of other agents leading to improved cooperation and coordination (Street, 2024). Based on these insights, there is a growing interest in leveraging ToM in LLMs to enhance multi-agent cooperation (Guo et al., 2023; Li et al., 2023b), where ToM plays a vital role in facilitating the coordination of actions and the resolution of conflicts in cooperation. Recent research has

highlighted the importance of accurate mutual understanding for effective cooperation in complex environments (Wang et al., 2022; Li et al., 2023a; Chan et al., 2023; Zhang et al., 2024a; Wu et al., 2024). This mutual understanding involves *comprehending other agents' profiles and trajectories*, resulting in enhanced coordination and cooperation across the multi-agent system. Following this perspective, we consider cooperative trends as the tendency of agents to exhibit accurate predictions about their teammates' actions.

To further identify the relationship between ToM ability and cooperative trend, we investigate the cooperative trend of agents with high and low ToM abilities. However, our result reveals that *agents with lower level ToM exhibited better cooperative trend compared to those with higher level ToM* (as detailed in Section 3). This suggests that *having a high level of ToM alone may not always lead to better cooperation*. Intuitively, agents with higher level ToM may overthink and anticipate potential conflicts, resulting in more cautious cooperation. This finding aligns with the psychological research by (Ridinger & McBride, 2017), which suggests that ToM capabilities alone are not sufficient to guarantee good cooperation. Instead, **agents may also need to be willing to positively reciprocate and cooperate with others**. Specifically, Ridinger & McBride (2017) highlights that when ToM abilities are combined with a high proportion of individuals who are willing to engage in cooperation, it can lead to improved cooperation within the group. This insight motivates our proposed approach of **incorporating belief alignment into the coalition formation process to promote cooperation among agents with high cognitive abilities**.

In this work, we propose a novel matching coalition mechanism to find coalitions that foster cooperation and leverage the strengths of agents by explicitly considering belief alignment as captured by their ToM capabilities. By forming coalitions with diverse ToM levels, we aim to create coalitions that can effectively reason about each other's mental states while leveraging their unique strengths to solve complex problems. Moreover, our matching algorithm allows for stable matching, which ensures the long-term viability and effectiveness of the formed coalitions.

**Contribution.** We summarized our main contributions as follows:

1. We investigate the **interplay between ToM capabilities and cooperative trend** in LLM-based multi-agent systems. Our study reveals agents with higher ToM capabilities may not necessarily exhibit better cooperative trends. These insights pose new challenges in understanding the complex interplay between cognitive abilities and cooperative trends, demonstrating the *potential of leveraging cognitive insights to design effective multi-agent mechanisms*.
2. We introduce a **stable coalition formation mechanism for team selection among LLM agents**, based on the alignment between agents' beliefs (derived from their ToM capabilities) and the actual actions of their potential partners. By establishing preference orders based on belief-action alignment, our mechanism forms effective coalitions and fosters cooperation among agents.
3. We conduct a comprehensive experimental evaluation incorporating ToM capabilities to assess the effectiveness of the proposed methods in facilitating cooperation, coalition stability, and task-specific performance.

## 2 RELATED WORK

**Agents with Theory of Mind (ToM).** The concept of ToM, which involves the ability to attribute mental states to oneself and others, has been studied extensively in cognitive science and psychology (Premack & Woodruff, 1978; Frith & Frith, 2003). In the field of multi-agent reinforcement learning, existing research mainly uses supervised training to equip agents with ToM capabilities for cooperative tasks (Wen et al., 2018; Wang et al., 2022; Oguntola et al., 2023).

With the rapid advancement of large language models (LLMs), some recent works have explored empowering LLMs with ToM capabilities (Arodi & Cheung, 2021; Li et al., 2022; Zhou et al., 2023; Xu et al., 2024). However, these efforts have primarily focused on developing ToM capabilities for individual agents, with limited exploration of how ToM can facilitate cooperation among multiple agents. While Li et al. (2023b) studied LLM agents with ToM capabilities and leveraged ToM capabilities specifically for cooperative tasks, the relationship between ToM ability and cooperative trend remains unclear.

In this work, we aim to explore the interplay between ToM ability and cooperative trend. Unlike previous studies that primarily focused on equipping individual agents with ToM capabilities, we investigate how different levels of ToM abilities influence cooperative trends in multi-agent systems.

**Multi-agent Cooperation.** Multi-agent cooperation has been studied extensively in both industry and academia, with traditional approaches focusing on game-theoretic frameworks (Shenoy, 1979;

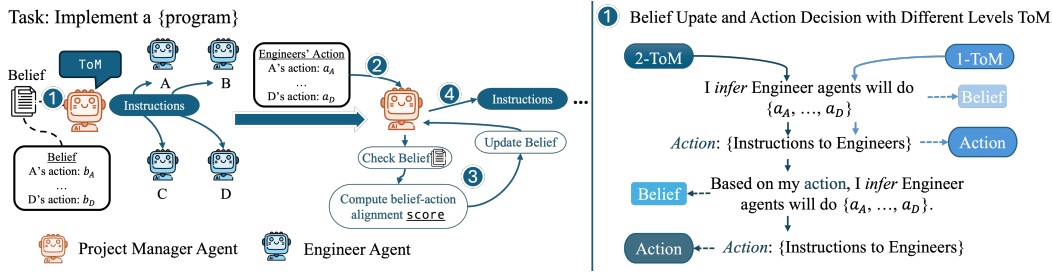

Figure 1: **Illustration for the multi-agent system setup and the ToM cognitive thinking process**. (*Left*) The multi-agent system setup for the iterative programming tasks, consisting of one project manager (PM) with ToM ability and four Engineers. The iterative process involves: ❶ PM updates its beliefs and takes actions based on its ToM reasoning; ❷ PM observes Engineers' actions; ❸ Evaluate cooperation by the alignment between PM's beliefs and Engineers' actions; ❹ PM provides instructions to Engineers. (*Right*) The ToM cognitive thinking process of PM agent involves recursive belief updates and decision-making based on the inferred beliefs.

Table 1: **Comparison of cooperative trends between agents with lower-level (k=1) and higher-level (k=2) ToM** in the Iterative Programming task on two benchmarks, including HUMANEVAL (Chen et al., 2021) and MBPP (Austin et al., 2021), over R = 1 . . . 5 interaction rounds: Low ToM agents show Higher cooperative trends.

| | $\text{ToM}^k$ | gpt-3.5-turbo | | GLM-4 | | Llama-3-70b | | Gemini-1.5-flash | | Claude-3-sonnet | |
| --- | --- | --- | --- | --- | --- | --- | --- | --- | --- | --- | --- |
| | | R=1 | R=5 | R=1 | R=5 | R=1 | R=5 | R=1 | R=5 | R=1 | R=5 |
| HUMANEVAL | k=1 | 62.5 | 51.7 | 65.5 | 63.3 | 80.9 | 75.0 | 75.0 | 84.72 | 67.8 | 67.1 |
| | k=2 | 50.0 | 48.0 | 63.2 | 60.8 | 75.0 | 73.5 | 80.56 | 80.56 | 63.8 | 57.2 |
| MBPP | k=1 | 44.3 | 35.8 | 83.1 | 85.2 | 81.3 | 85.3 | 65.74 | 66.67 | 57.6 | 48.6 |
| | k=2 | 31.3 | 35.8 | 82.0 | 86.3 | 81.7 | 82.6 | 60.58 | 66.67 | 52.8 | 54.4 |

Yoshida et al., 2008) and negotiation and communication strategies (Foerster et al., 2016; Tang, 2019; Yang et al., 2024), enabling agents to learn cooperative strategies through interaction and information exchange. The advancements of LLMs have opened up new opportunities for multi-agent cooperation, leveraging their capabilities in natural language understanding, generation, and reasoning (Zhang et al., 2024b; Talebirad & Nadiri, 2023). Recent works have explored utilizing LLM agents to coordinate and cooperate in various tasks and frameworks, *e.g.*, CAMEL (Li et al., 2023a) employs role-play to facilitate autonomous cooperation, AutoGen (Wu et al., 2023a) uses conversable agents and conversation programming, and MetaGPT (Hong et al., 2024) incorporates human workflows into LLM-based multi-agent collaborations to ensure more coherent and effective teamwork. In addition, recent research has focused on improving specific aspects of multi-agent collaboration, such as conversation for knowledge sharing (Xu et al., 2023; Lan et al., 2023; Hua et al., 2023; Wu et al., 2023b; Nascimento et al., 2023; Fu et al., 2023) and cognitive ability (Liang et al., 2023; Gong et al., 2023), demonstrating the potential of leveraging language to facilitate cooperation and coordination. For effective cooperation in complex and dynamic environments, some studies focus on the dynamic adjustment of group members and the selection of teammates to improve cooperation and overall performance (Chen et al., 2024; Li et al., 2023a; Shi et al., 2023). Besides, Liu et al. (2024) proposes the Dynamic LLM-Agent Network (DyLAN) framework to optimize team performance based on task queries and peer ranking.

Different from previous work on optimizing team members, our work mainly focuses on the interplay of cognitive abilities and cooperative trends in multi-agent systems. We underscore the importance of fostering effective cooperation by considering cognitive aspects.

## 3 MOTIVATION

Theory of Mind (ToM) in multi-agent cooperation has been studied recently and agents with higher ToM capabilities are generally expected to be more effective in understanding and predicting the actions of others, which leads to improved cooperation and coordination. However, the relationship between ToM ability and cooperative trend in multi-agent systems is not fully understood, and there are still open questions regarding the impact of ToM ability on cooperation.

In this study, we employ a standard ToM model aligned with current research (Zhou et al., 2023; Street, 2024; Xu et al., 2024) to understand ToM in multi-agent cooperation. In practice, ToM

typically does not exceed second-order reasoning due to cognitive limitations and diminishing returns (Premack & Woodruff, 1978; Frith & Frith, 2003). We consider two levels of ToM ability, aligning with existing literature in both human cognitive research (De Weerd et al., 2015) and agent cognitive modeling (Li et al., 2023c): Low ToM (Level 1), where agents can represent and reason about others' beliefs, desires, and intentions; and High ToM (Level 2), where agents can additionally consider others' ToM reasoning.

**Higher-level ToM capabilities do not necessarily guarantee better cooperative trends**. Accurate mutual understanding is crucial for effective cooperation in complex multi-agent environments. Recent research has explored communication for knowledge sharing (Wang et al., 2022; Li et al., 2023a) and aligning agent beliefs with teammates (Chan et al., 2023; Zhang et al., 2024a; Wu et al., 2024), demonstrating that improved alignment can facilitate more informed decision-making and potentially lead to more cooperative outcomes. Motivated by these insights, we introduce the *"Fraction of Trusted Members" (FTM)* metric to quantify the cooperative trend (detailed in Section 6). FTM measures the alignment between the Project Manager's (PM's) beliefs about the engineer agents' actions and their actual actions, representing the proportion of engineer agents whose actions are correctly anticipated by the PM. Additionally, we present comprehensive evaluations across multiple performance metrics and tasks in Appendix F, which further support our findings.

To investigate the relationship between ToM and cooperative trend in multi-agent systems, we first simulate the multi-agent system consisting of one project manager (PM) and four engineer agents working together to solve a programming task, as shown in Figure 1 (*Left*). Specifically, PM is enabled with ToM ability, which means he can recursively infer the actions of other agents (as beliefs) and take his own actions accordingly, as illustrated Figure 1 (*Right*). Then, we investigate the multi-agent cooperation with iterative programming tasks (five rounds) on two benchmarks, HUMANEVAL (Chen et al., 2021) and MBPP (Austin et al., 2021). In Appendix F, we also investigate side effects of incorporating higher ToM in some state-of-art methods.

Notably, our results revealed that *a low ToM agent (k=1) exhibited better cooperative trend compared to a high (k=2) ToM agent*, as detailed in Table 1. This suggests that agents with high ToM capabilities may overthink and anticipate potential conflicts, resulting in more cautious cooperation. To leverage the strengths of agents with different ToM levels, we propose a coalition matching algorithm to foster cooperation and improve cooperative task performance.

The rest of this paper is organized as follows: Section 4 introduces the ToM modeling and the matching problem in multi-agent systems. Section 5 describes our proposed coalition matching algorithm to foster cooperation among agents with diverse ToM capabilities. Section 6 presents the experiments evaluating the impact of ToM and the matching algorithm on cooperative trend.

## 4 FORMULATION

In this section, we present the novel formulation for incorporating ToM capabilities and stable matching theory into multi-agent LLM cooperation. In multi-agent cooperation scenarios, agents typically have defined roles and goals. We leverage the alignment between agents' beliefs and actions from ToM to establish coalitions, fostering more stable collaborations.

### 4.1 ToM FORMULATION FOR LLM AGENTS

Theory of Mind refers to the ability to attribute mental states, such as beliefs, intentions, and desires, to oneself and others, and to understand that others may have beliefs and intentions that differ from one's own. In the context of LLM-empowered multi-agent systems, for an LLM agent $i$ at interaction round $R$, we define its $k$-level ToM function as:

$$\text{ToM}_i^k(o_i^{1:R}, \hat{a}_{-i}^{1:R-1}, \{b_{i,R}^{k-1}(a_m^R)\}_{m \neq i}) := b_{i,R}^k, \tag{1}$$

Where:
- $o_i^{1:R}$ represents agent $i$'s observation history up to round $R$, including current task state, self actions, and collaborate teammates.
- $\hat{a}_{-i}^{1:R-1}$ represents other agents' action history up to round $R-1$.
- $\{b_{i,R}^{k-1}(a_m^R)\}_{m \neq i}$ captures agent $i$'s prediction of agent $m$'s action at round $R$ based on $(k-1)$-level ToM reasoning: $b_{i,R}^{k-1}(a_m^R) = p(a_m^R | \text{ToM}_i^{k-1}(o_i^{1:R}, \hat{a}_{-i}^{1:R-1}, \{b_{i,R}^{k-2}(a_l^R)\}_{l \neq i}))$.

Specifically, $b_{i,R}^k$ represents agent $i$'s nested beliefs at level $k$ in round $R$, captures the agent 's belief about other agents at the corresponding level of recursion $k-1, \ldots 0$.

For the base case of 0-level ToM, $b_{i,R}^0$ just record cooperation history $(o_i^{1:R}, \hat{a}_{-i}^{1:R-1})$, without considering any ToM reasoning: $b_{i,R}^0 = \text{ToM}_i^0(o_i^{1:R}, \hat{a}_{-i}^{1:R-1})$.

**Remarks**: These LLM-empowered agents operate in a vast, open-ended action space defined by natural language (Gur et al., 2023), presenting the "observation" ($o_i^{1:R}$), "actions" ($\hat{a}_{-i}^{1:R-1}$) and "beliefs" ($b_{i,R}^k$) as *textual outputs*. We provide carefully designed prompts to define each agent's role and level of ToM, guiding the generation of ToM-based reasoning and facilitating each agent's belief updates (see Appendix C for detailed examples).

### 4.2 Matching Formulation for Multi-Agent LLMs

Considering the set of LLM agents $N = \{1, 2, \ldots, n\}$, and a matching $\mu$ assigns each agent to a coalition such that $|\mu(i)| \geq n - 1$ for all $i \in N$, where $n$ is the minimum coalition size (typically set to $\lceil N/2 \rceil$ in our experiments). Each agent $i$ has preferences over potential coalitions $S \subseteq N \setminus i$ based on the average belief-action alignment score:

$$B_i(S) = \frac{1}{|S|} \sum_{j \in S} \phi(b_i^k(a_j) - \hat{a}_j), \tag{2}$$

Where $b_i^k$ represents agent $i$'s $k$-level belief, $\hat{a}_j$ is agent $j$'s actual action, and $\phi$ is an alignment measure[1].

For two potential coalitions $S_1, S_2$: $S_1 \succ_i S_2 \Leftrightarrow B_i(S_1) < B_i(S_2)$. A matching $\mu$ is stable if there exists no blocking coalition $C \subseteq N$ such that: (a) $|C| \geq n - 1$ (minimum size requirement), and (b) $\forall i \in C : C \succ_i \mu(i)$ (coalition preferred by all members).

Additional, introduce the tolerance parameter $\epsilon$ to reduce the search space by filtering out poorly aligned agent pairs. Importantly, we maintain robustness even when all pairwise alignment scores exceed $\epsilon$. In such cases of universal misalignment, the preference order remains well-defined through the coalition scoring function $B_i(S)$. This ensures robust coalition formation even in challenging scenarios: *agents still form coalitions of minimum size $n$ with their relatively best-aligned partners* based on $B_i(S)$ scores. Formally, while $\epsilon$ helps computational efficience ($\phi(b_i^k(a_j) - \hat{a}_j) \leq \epsilon$), the *stability conditions and preference ordering remain valid* even when this constraint is relaxed.

**Remarks**: The alignment between beliefs and actions is not a mathematical subtraction, but rather a measure of semantic similarity or alignment. To calculate this alignment score, we employ a self-evaluation approach involving prompting the agent to evaluate the alignment between its belief and another agent's action, which is consistent with existing LLM agent literature (Qin et al., 2023; Zheng et al., 2023; Liu et al., 2024).

The stable matching problem for multi-agent LLMs can then be formulated as finding a matching $\mu$ that satisfies the stability condition, given the agents' preferences based on their $k$-level ToM beliefs and the alignment between their beliefs and the actual actions of other agents. By capturing the agents' preferences based on the alignment between their beliefs and the actual actions of other agents, we can investigate how the agents' ToM ability impacts their cooperative trend and the overall stability of the multi-agent system.

## 5 Cooperation Mechanism for Multi-Agent LLMs

In this section, we propose a cooperation mechanism for multi-agent LLMs that enables agents to form and maintain stable cooperative coalitions by leveraging their ToM functions to predict and align their beliefs with their partners' actions. To further enhance the effectiveness of the coalition, we introduce a specialized ability-matching adaptation that prioritizes agents with crucial skills for specific tasks. This adaptation optimizes the formation of coalitions, ensuring agents possessing the necessary expertise are included in the coalition.

### 5.1 Coalition Matching Mechanism for Multi-Agent LLMs

To foster the cooperation among agents with different ToM levels, we introduce the multi-agent LLM cooperation mechanism operates in cooperation rounds, where agents form stable cooperative coalitions for team selection, detailed in Algorithm 1. In each round, every agent $i \in N$ uses its

---

[1]In this paper, we employ a self-evaluation approach involving prompting the agent to evaluate, exemplified in Appendix A. Besides, we also discuss belief-alignment calculation for non-LLM agents in Appendix A

---

**Algorithm 1 Multi-Agent Coalition Matching Mechanism**

---

**Require:** $\mathcal{N} = \{1, 2, \ldots, n\}$: the set of LLM agents; $k$: the desired level of recursion for ToM; $\epsilon \geq 0$: error tolerance for belief alignment.

1: Initalize $S \leftarrow \mathcal{N}$; `rematching_required = -1`;
2: **for** each cooperation round $R$ **do**
3:    **for** each agent $i \in N$ **do**
4:       $b_{i,R}^k \leftarrow \text{ToM}_i^k(o_i^{1:R}, \hat{a}_{-i}^{1:R-1}, \{b_{i,R}^{k-1}(a_m^R)\}_{m \neq i})$ {Belief Update}
5:    **end for**
6:    **if** $R = 1$ or `rematching_required = 1`:
7:       Establish preference order $\succ_i$ based on $b_{i,R}^k$ and $\epsilon$ {Preference Ordering Equation (2)}
8:       Update stable coalition $S$ based on preference orders $\{\succ_i\}$ {Stable Matching}
9:       `rematching_required = -1`;
10:    **for** each agent $i \in S$ **do**
11:       Cooperate with assigned partner $\mu(i) = S \setminus \{i\}$ {Coalition Formation & Task Execution}
12:       **if** $\phi\left(b_{i,R}^k(a_j^R) - \hat{a}_j^R\right) > \epsilon$ for $j \in \mu(i)$: Signal desire to re-match
13:    **end for**
14:    **if** environment has re-matching: `rematching_required +1`;
15: **end for**

---

$k$-level ToM function $\text{ToM}_i^k(\cdot)$ to form beliefs $b_i^k$ about the mental states of other agents, based on its observations $o_i$, the actions $\hat{a}_{-i}$ of others, and the $(k-1)$-level beliefs $b_{-i}^{k-1}$ of others. Then, agent $i$ establishes a preference order $\succ_i$ over potential partners based on the alignment between its belief $b_i^k(a_j)$ and agent $j$'s actual action $\hat{a}_j$ within a tolerance $\epsilon$. The agents form coalitions by cooperating with their assigned partners $\mu(i)$ in the stable matching. The optimal coalition $S$ with the strongest belief alignment will be the team selected for cooperation (Algorithm 1, Line 3~11).

Then, if the alignment between an agent $i$'s belief $b_i^k(a_{\mu(i)})$ and its partner's action $\hat{a}_{\mu(i)}$ falls below the tolerance $\epsilon$, the agent signals a desire to re-match, triggering new cooperation round with a stable matching computation (Algorithm 1, Line 12). This iterative process allows agents to form and maintain stable coalitions while adapting to changes in beliefs and preferences over time, leveraging their ToM capabilities, belief alignment, and stable matching principles.

***Remarks***: For $k$-level ToM agent $i$'s action, it can be represented by $\hat{a}_i = f(b_i^k, \mu(i))$, where $f$ is the LLM agent's decision-making process. The action $\hat{a}_i$ thus encapsulates the complex interplay between the agent's belief $b_i^k$ and partners $\mu(i)$ in the stable matching on its decision-making process. For rematching, our algorithm includes an *adaptation* phase rather than immediate reformation. When agents signal misalignment, the current coalition continues for one additional round, allowing agents to *observe and adapt* to rematch signals.

## 5.2 ADAPTATION FOR SPECIALIZED AGENT ABILITIES

While the proposed mechanism for multi-agent LLM cooperation focuses on the alignment between agents' beliefs and actions, it is also important to consider the specialized abilities of individual agents, especially when forming coalitions for cooperative tasks that require specific skills or capabilities.

In scenarios where certain agents possess specialized abilities that are highly relevant to the cooperative task at hand, the original stable matching algorithm may not necessarily prioritize these agents, as it solely relies on the alignment between beliefs and actions. To address this limitation, we propose an adaptation to the mechanism that incorporates agents' specialized abilities into the matching process.

Let $\alpha_i$ represent the specialized ability score of agent $i$ for the cooperative task under consideration. Higher values of $\alpha_i$ indicate greater specialized ability for the task. We can modify the coalition preference order $\succ_i$ of each agent $i$ to incorporate both belief alignment and specialized abilities. The modified preference score for a coalition $S$ is defined as: $B_i'(S) = B_i(S) + \lambda \cdot \frac{1}{|S|} \sum_{j \in S} \alpha_j$.

where $\lambda$ is a weighting parameter (default is 1 in our evaluation). The updated preference order $\succ_i'$ between coalitions $S_1$ and $S_2$ is then defined as:

$$S_1 \succ_i' S_2 \iff B_i'(S_1) < B_i'(S_2)$$

***Remarks***: This formulation maintains transitivity in coalition preferences while balancing cognitive alignment ($B_i(S)$) with task-specific capabilities ($\{\alpha_i\}_{i \in S}$). The coalitions now form based on both belief-action alignment and specialized abilities. Importantly, the preference structure remains

well-defined for coalitions of different sizes meeting the minimum requirement $n$. The detailed proofs for convergence, stability, and cycle-freedom with specialized abilities are provided in Appendix G.

By incorporating the specialized ability scores $\alpha_i$ into the preference order, the stable matching algorithm will prioritize agents with higher specialized abilities for the cooperative task and consider the belief alignment. This adaptation ensures that agents with crucial specialized abilities are more likely to be included in the formed coalitions, enhancing the overall effectiveness of the cooperation mechanism for tasks that require specific skills or capabilities (as discussed in Appendix C.1).

## 6 EXPERIMENT

### 6.1 EXPERIMENT SETUP

The experiment setup involves extending the MetaGPT framework (Hong et al., 2024) to incorporate the proposed multi-agent LLM cooperation mechanism. LLM agents will have varying levels of ToM capabilities, including 1-level, and 2-level ToM. We evaluate our proposed coalition mechanism on the following cooperative tasks:

- *Iterative Programming:* In this task, agents simulate a software development team, with each agent assuming different roles. We use HUMANEVAL (Chen et al., 2021) and MBPP (Austin et al., 2021) as benchmarks to evaluate the performance and cooperative trend of ToM agents in this multi-agent environment.
- *Debate:* In a debate setting, agents are divided into two sides (affirmative and negative) and engage in a multi-round debate on a given topic. The agents collaborate within their respective sides to present arguments and counterarguments.
- *Logical and General Reasoning:* We evaluate the effectiveness of our coalition matching mechanism in reasoning tasks using the AQUA-RAT dataset (Ling et al., 2017, Logic Reasoning) and MMLU dataset (Hendrycks et al., 2021, General Reasoning), where agents collaborate to solve complex reasoning questions.

**Models.** We utilize five state-of-the-art LLMs in our multi-agent cooperation scenarios: GPT3.5 (OpenAI, 2023) (`gpt-3.5-turbo`), GLM (`GLM-4`), Llama 3 (Meta AI, 2024) (`Llama-3-70b`), Gemini (Velloso & Woodward, 2024) (`Gemini-1.5-flash`), and Claude (Templeton et al., 2024) (`Claude-3-sonnet`).

### 6.2 EVALUATION METRICS

To comprehensively evaluate the effectiveness of the proposed multi-agent LLM cooperation mechanism, we define the following metrics:

**Fraction of Trusted Members (FTM)**: This metric evaluates the cooperative trend of the ToM agent based on the Belief-Action Alignment score. We first define a threshold $\epsilon$ for the alignment score; if an agent's score is below this threshold, the agent is considered a trusted member. An agent $j$ is a trust member for agent $i$ with $k$-level ToM if the belief-action alignment score $A_{i,j}^k \leq \epsilon$. The FTM for agent $i$ with $k$-level ToM, denoted as $\text{FTM}_i^k$, is then calculated as the fraction of Trusted Members among all other agents: $\text{FTM}_i^k = \frac{1}{n-1} \sum_{j \neq i} \mathbb{1}(A_{i,j}^k \leq \epsilon)$ where $\mathbb{1}(\cdot)$ is the indicator function, and $A_{i,j}^k$ is the belief-action alignment score between agent $i$ with $k$-level ToM and agent $j$, computed as: $A_i^k = \frac{1}{n-1} \sum_{j \neq i} \phi\left(b_i^k(a_j) - a_j\right)$, where $b_i^k(a_j)$ represents agent $i$'s belief about agent $j$, $a_j$ is agent $j$'s actual action, and $\phi(\cdot)$ is alignment score evaluated by agent itself (demonstrated in Appendix A). A higher FTM value indicates better cooperation trends, as it reflects a more accurate comprehension of other agents' profiles and trajectories (discussed in Section 1).

**Coalition Stability with ToM**: We measure the stability of coalitions formed by agents with ToM capabilities by considering the average lifetime of these coalitions. The coalition lifetime is defined as the number of cooperation rounds that a coalition remains stable before a re-matching event occurs due to belief-action misalignment exceeding a predefined tolerance threshold $\epsilon$. To calculate the average coalition lifetime, we use $\frac{1}{m} \sum_{c=1}^{m} l_c$, where $m$ represents the number of times matching is triggered throughout the cooperation rounds, and $l_c$ denotes the lifetime of the coalitions formed after the $c$-th matching event, where the initial coalition $l_1$ includes all agents.

### 6.3 MAIN RESULTS

**Cooperative Trend with ToM ability**: To investigate the impact of ToM on cooperative trend in multi-agent systems, we evaluate the Iterative Programming task on two benchmarks, including

Table 2: **Comparison of cooperative trends between agents with lower-level ToM (1-level) and higher-level ToM (2-level)** in the Iterative Programming task ( HUMANEVAL ) over 5 interaction rounds: Without matching, low ToM agents show higher cooperative trends; While with matching stable coalition, high ToM agents achieve higher cooperation ability as cooperation progresses.

| | | gpt-3.5-turbo | | GLM-4 | | Llama-3-70b | | Gemini-1.5-flash | | Claude-3-sonnet | |
| | | 1st Rnd. | 5th Rnd. | 1st Rnd. | 5th Rnd. | 1st Rnd. | 5th Rnd. | 1st Rnd. | 5th Rnd. | 1st Rnd. | 5th Rnd. |
|---|---|---|---|---|---|---|---|---|---|---|---|
| w.o. | 1-ToM | 62.5 | 51.75 | 65.5 | 63.25 | 80.88 | 75.00 | 75.0 | 84.72 | 67.76 | 67.11 |
| Matching | 2-ToM | 50 | 48 | 63.25 | 60.75 | 75.00 | 73.53 | 80.56 | 80.56 | 63.82 | 57.24 |
| w. Matching | 1-ToM | **81** | 77.75 | 83.25 | 84.0 | **95.83** | 97.50 | **88.89** | 89.29 | **94.23** | 91.07 |
| (Ours) | 2-ToM | 76.25 | **80.75** | 83.25 | **91.0** | 93.18 | **97.50** | 88.64 | **95.00** | 93.75 | **95.45** |

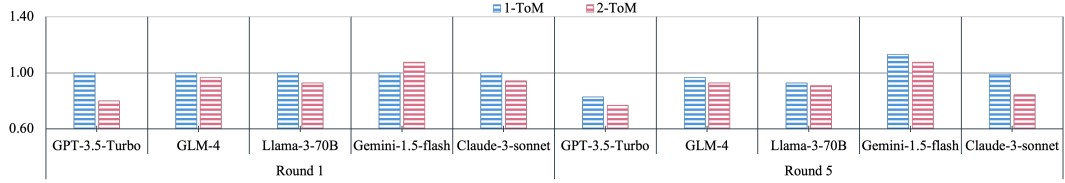

(a) **w.o. Matching**: 1) Agents with *low ToM ability* exhibited a *higher cooperation trend* compared to high ToM agents; 2) ToM agents demonstrate a *decline in cooperative trend* as the collaboration progresses.

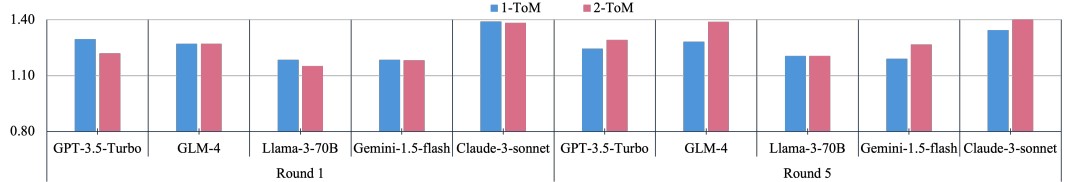

(b) **w. Stable Matching (Ours)**: 1) *The matching mechanism leads to higher cooperative trend* for both low and high ToM agents compared to the corresponding values in the "w.o. Matching" setting in Figure 2a; 2) With matching stable coalition, *high ToM agents achieve higher cooperation ability* as cooperation progresses.

Figure 2: **Comparison of cooperative trends between agents with low (1-ToM) and high (2-ToM) abilities** in the Iterative Programming task ( HUMANEVAL ) over 5 collaboration rounds under "w.o. Matching" and "w. Stable Matching" settings.

HUMANEVAL and MBPP. The multi-agent system consists of one Project Manager (PM) and four Engineers for task execution in the baseline without matching.

We vary PM's ToM ability between 1-level (low) and 2-level (high) to observe the effects on coalition formation and cooperative trend. Based on the belief-action alignment, PM will select coalition members as described in the proposed cooperation matching mechanism (Algorithm 1). To quantitatively assess the impact of ToM ability on cooperative trend, we measure the Fraction of Team Matching (FTM) for the ToM agent, with a higher FTM value indicating a more cooperative agent. Table 2 presents the results of our experiments on the HUMANEVAL benchmark, comparing the cooperative trend of agents with lower-level (1-level) and higher-level (2-level) ToM in two settings: without a matching coalition and with a matching coalition formed using our proposed cooperation mechanism. Similar results for the MBPP benchmark are provided in Appendix B.

To facilitate comparison across different models, we first calculate the FTM value for each model's 1-ToM agent in the 1st round of the no-matching setting. This value serves as a baseline for different ToM levels and matching/no-matching conditions. Then, we divide the obtained FTM values by the corresponding model's baseline FTM value. For example, all values corresponding to GLM-4 in Table 2 should be divided by 65.5. The normalized results are illustrated in Figure 2.

**In the absence of a matching coalition (Figure 2a), agents with *low ToM ability* exhibited a *higher cooperation trend*** compared to agents with high ToM ability over 5 collaboration rounds. This suggests that agents with high level ToM may be less likely to cooperate, possibly due to their tendency to overthink and anticipate potential conflicts, leading to more cautious cooperation.

When the matching coalition is formed, we observed an increase in cooperative trend for both low and high ToM ability agents as shown in Figure 2b. This demonstrates the effectiveness of our proposed cooperation mechanism in promoting cooperation among agents with ToM. Interestingly, **in the coalition setting (Figure 2b), agents with *high ToM ability* show a *higher cooperation ability* compared to low ToM ability agents as collaboration progressed.** By the end of the 5th round, high ToM ability agents exhibited a higher cooperation rate than low ToM ability agents.

Table 3: **Comparison of the coalition stability and Pass@1 performance of MetaGPT and our proposed approach** with 1-ToM and 2-ToM agents using stable matching on the HUMANEVAL and MBPP benchmarks for the Iterative Programming task. The coalition stability is measured by the average number of rounds the formed coalitions remain stable out of the total 5 rounds.

| | HUMANEVAL | | MBPP | |
| --- | --- | --- | --- | --- |
| | Coalition Stability | Pass@1 | Coalition Stability | Pass@1 |
| MetaGPT | – | 85.4% | – | 86.5% |
| 1-ToM w. Matching | 3.4/5 | 87.2% | 3.7/5 | 88.2% |
| 2-ToM w. Matching | **3.6/5** | **90.0%** | **4.0/5** | **90.4%** |

This indicates that when the matching algorithm is introduced, it provides a stable coalition for cooperation, where high ToM ability agents can fully utilize their perspective-taking skills to maintain and enhance cooperation over time.

**Coalition Stability with ToM**: To evaluate the effectiveness of our proposed stable matching approach, we compare the task performance of MetaGPT and our method with 1-ToM and 2-ToM agents on two Iterative Programming benchmarks: HUMANEVAL and MBPP. To assess the performance of the approaches on the programming task, we use the Pass@1 metric (Hong et al., 2024), which represents the percentage of test cases passed by the generated code on its first attempt. A higher Pass@1 score indicates better code quality and problem-solving ability. Moreover, Moreover, we measure coalition stability for multi-agent teams with a ToM agent (PM) by calculating the average number of rounds the formed coalitions remain stable out of 5 rounds. A coalition is stable if no agent in the coalition desires to leave and form a new coalition based on their belief-action alignment.

As shown in Table 3, our approach with 1-ToM and 2-ToM agents using stable matching achieves higher Pass@1 scores and maintains more stable coalitions compared to MetaGPT on both the HUMANEVAL and MBPP benchmarks. The 2-ToM agents with stable matching demonstrate the best performance, maintaining stable coalitions for an average of 3.6 out of 5 rounds on HUMANEVAL and 4.0 out of 5 rounds on MBPP, while also achieving *the highest Pass@1 scores* of 90.0% and 90.4%, respectively. These results highlight the improved cooperation ability of high ToM agents in maintaining stable coalitions over time, which contributes to better task performance. By incorporating ToM and stable matching, our proposed approach enables agents to form stable and high-performing coalitions, leading to enhanced collaboration and task performance.

### 6.4 CASE STUDY: DEBATE

To evaluate the effectiveness of our proposed coalition matching mechanism, we designed a debate task using the MetaGPT framework (Hong et al., 2024). In this task, agents (`gpt-4-0613`) are divided into two sides: the affirmative side and the negative side, with each side comprising three debater agents. Each side's objective is to cooperate and win the debate.

At the beginning of our investigation, we explore the cooperative trends of low and high ToM agents in a debate setting with the topic "Should the Death Penalty be abolished?" (Roush & Balaji, 2020; Baturo et al., 2017), and both low and high ToM agents are assigned to the affirmative side. The following example presents the thinking and actions of debaters with different levels of ToM, as detailed in Appendix D. We observe that higher ToM agent acts as less cooperative, which confirms our previous analysis.

---

**Compare Cooperative Trends between low and high ToM agents**

**Thinking and Action of 1-ToM Debater**: I anticipate my teammates making these common arguments against the death penalty. *To complement their points*, I will focus my argument on the lack of evidence supporting the death penalty's effectiveness as a crime deterrent. I will argue that there is no conclusive evidence showing that the death penalty deters crime more effectively than lengthy imprisonment, making it an unnecessary and unjustified punishment. ⇐ Focuses on complementing their teammates' points & Good Cooperative Trends

**Thinking and Action of 2-ToM Debater**: My arguments about deterrence and discrimination will prompt my teammates to make related points to strengthen the overall case. *I should use my next arguments to introduce new issues*. I will argue that the death penalty often involves inhumane and botched executions, and that most democracies and U.S. states have abolished or prohibited it. ⇐ Overthinking & Worse Cooperative Trend

---

Then, we conducted experiments under three different settings to show the effectiveness of our matching mechanism for fostering cooperation: (1) *No-ToM (Baseline)*: Each side consists of 3

Table 5: Comparative Evaluations for Logic Problem-Solving and General Reasoning

| | Logic Problem-Solving | | General Reasoning | |
|---|---|---|---|---|
| | gpt-3.5-turbo Acc (%) | gpt-4o-mini Acc (%) | gpt-3.5-turbo Acc (%) | gpt-4o-mini Acc (%) |
| ChatEval w. ToM | 40.23 | 69.14 | 54.39 | 69.90 |
| DyLAN w. ToM | 43.50 | 68.50 | 57.92 | 72.98 |
| Ours (+matching) | **45.70** | **75.39** | **60.94** | **75.57** |

debater agents without ToM. For each speech, two debaters were randomly selected from each side; (2) *ToM without Matching*: The three debaters on the affirmative side have varying ToM levels (0-level, 1-level, and 2-level), while the negative side had no ToM. For each speech, two debaters were *randomly selected* from each side. This setting allows us to evaluate the impact of having an affirmative team with ToM against a team without ToM; (3) *ToM with Matching*: Similar to the ToM without Matching setting, the affirmative side has three debaters with varying ToM levels. However, the affirmative side selects two debaters using our proposed coalition matching mechanism.

**Foster Coalition Matching for agents with Varying Order ToM**. For the debate topic "Should the Death Penalty be abolished?", the debate consists of 5 rounds, with each side alternating to speak. We conducted the debate 11 times, and the outcomes were evaluated by gpt-4-0613. As shown in Table 4, for the tested side ToM settings use varied ToM levels while keeping the opposing side as No-ToM agents. When testing the affirmative side (Aff.), matching improves Aff.'s win rate of 67.27%, outperforming both the No-ToM setting (65.45%) and the ToM baseline without matching (61.82%). Similarly, when testing the negative side (Neg.) with matching, the win rate surpasses the other two settings.

Table 4: **Win rates for both debate sides**.

| Setting | Aff. | Neg. |
|---|---|---|
| No-ToM | 65.45% | 34.55% |
| ToM w.o. Matching | 61.82% | 25.45% |
| ToM w. Matching (Ours) | **67.27%** | **36.36%** |

Besides, for the coalition matching among agents with different ToM levels, our experiments show that the *initial* cooperation rate between the 1-ToM and 2-ToM agents was relatively low, suggesting *higher-level ToM may not naturally form coalitions with each other*. Notably, with our coalition matching mechanism, the affirmative side's cooperation rate between the 1-ToM and 2-ToM agents increased from 9.1% (Round 1) to 18.2% (Round 5) as the debate rounds progressed. This suggests that our proposed method effectively fosters more effective cooperation among agents with ToM over interaction.

## 6.5 Evaluations on Reasoning Tasks

To demonstrate the effectiveness of our matching mechanism, we conducted comprehensive evaluations on two types of reasoning tasks: logic problem solving (using the AQUA_RAT dataset) and general reasoning (sampled from 4 subjects in MMLU dataset: "us_foreign_policy", "human_sexuality", "international_law" and "abstract_algebra"). We compared our approach against existing frameworks (ChatEval and DyLAN) with ToM capabilities. In Appendix E, we also demonstrate one comprehensive example to show how coalition works for fostering cooperation.

As shown in Table 5, our proposed matching mechanism consistently outperforms both ChatEval and DyLAN across all scenarios. These results demonstrate that our matching mechanism effectively leverages the strengths of agents with different ToM levels, fostering more effective multi-agent cooperation and improving performance.

## 7 Limiation, Future Work, and Conclusion

**Limitations and Future Work .** In this paper, we propose a novel coalition matching for fostering cooperation among LLM-based agents with ToM, demonstrating its effectiveness in enhancing cooperative trends. However, it is important to acknowledge the limitations of our approach. Firstly, LLMs may have limited reasoning ability despite their impressive language capabilities. Additionally, coalition formation is generally an NP-hard problem, and future research can explore potential optimizations to reduce time complexity. Furthermore, future research should investigate the incorporation of additional cognitive architectures, *e.g.* more advanced ToM models.

**Conclusion.** In this work, we investigated the relationship between cognitive abilities (ToM) and cooperative trends in LLM-based multi-agent systems. Our findings suggest that the relationship between ToM and cooperation is not always straightforward. Furthermore, we proposed a novel matching coalition mechanism incorporating cognitive insights into the design of multi-agent systems. By incorporating cognitive insights into the design of coordination mechanisms, we pave the way for exploring the translation of cognitive abilities into cooperative actions in multi-agent settings.

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

## A   AGENT SELF-EVALUATION FOR BELIEF ALGINMENT

Consider the 1-ToM Project Manganer, who reasoning the other Engineers' actions and trajectories and updates the belief as: "The engineers will need to understand the rules and mechanics of the 2048 game, design the game logic and user interface, and implement the code accordingly."

In the next interaction round, the Engineers provide the following actions:

```
actions = {
    "Engineer1": "Implemented using a 2D array to represent the
    ↪  game board, with functions for merging tiles and
    ↪  generating new tiles.",

    "Engineer2": "Used object-oriented approach with classes for
    ↪  Board and Tile, implementing game rules as methods."
}
```

Then, the Project Manager evaluate the belief alignment according to the following prompt:

---

**Instruction:**

**Prompts for Belief Alignment Evaluation**:
You will provide your belief alignment scores for each engineer's implementation based on your belief model.

```
Engineer's actions: {
    "Engineer1": "Implemented using a 2D array to represent
    ↪  the game board, with functions for merging tiles and
    ↪  generating new tiles.",

    "Engineer2": "Used object-oriented approach with classes
    ↪  for Board and Tile, implementing game rules as
    ↪  methods."
}
```

# Your belief model:
The engineers will need to understand the rules and mechanics of the 2048 game, design the game logic and user interface, and implement the code accordingly.
# Instruction:
For each engineer, provide a belief alignment score between -1 and 1. Respond in the following JSON format:

```
    {
    "Engineer1": {
        "score": float (-1 to 1). Belief alignment score for
        ↪  Engineer1,
        "explanation": Brief explanation of Engineer1's
        ↪  alignment score in 10 words or less,
        "justification": Detailed justification of
        ↪  Engineer1's alignment score, considering their
        ↪  implementation and your belief model, in 30-50
        ↪  words
    },
    "Engineer2": {
        "score": float (-1 to 1). Belief alignment score for
        ↪  Engineer2,
        "explanation": Brief explanation of Engineer2's
        ↪  alignment score in 10 words or less,
        "justification": Detailed justification of
        ↪  Engineer2's alignment score, considering their
        ↪  implementation and your belief model, in 30-50
        ↪  words
    }
    }
```

---

```
Outputs: Alignment Score

{
    "Engineer1": {
        "score": 0.7,
        "explanation": "Good game logic, basic UI, aligns
        ↪   with core requirements.",
        "justification": "Engineer1 demonstrates
        ↪   understanding of 2048 mechanics with a functional
        ↪   2D array implementation. The CLI interface, while
        ↪   basic, fulfills the requirement of implementing a
        ↪   user interface. However, there's room for
        ↪   improvement in UI design."
    },
    "Engineer2": {
        "score": 0.9,
        "explanation": "Excellent OOP approach, advanced UI
        ↪   with animations.",
        "justification": "Engineer2 shows a deep
        ↪   understanding of the game's rules and mechanics
        ↪   through their object-oriented design. The
        ↪   graphical interface with animations goes beyond
        ↪   the basic requirements, demonstrating strong
        ↪   skills in both game logic and UI implementation."
    }
}
```

**Remarks**. For LLM agents, our evaluation relies on semantic similarity analysis of natural language outputs. While we acknowledge the inherent limitations of LLM-based measurements, our coalition formation mechanism is a plug-and-play approach for improving multi-agent cooperation rather than advancing LLM evaluation methods.

Besides, we provide an extension of our framework to non-LLM agents where belief-action alignment can be more precisely quantified. For traditional reinforcement learning or robotic agents, we can define the belief-action alignment using *structured trajectories* $\tau = \{(s_t, a_t)\}_{t=1}^T$, where $s_t$ and $a_t$ represent states and actions at time step $t$. The alignment score between agent $i$'s belief $b_i^k(\tau_j)$ about agent $j$'s trajectory and $j$'s actual trajectory $\tau_j$ can be computed using established trajectory similarity metrics. Specifically, we can embed the state-action pairs using *domain-specific* feature extractors $\phi(s, a)$ and measure alignment through cosine similarity: $A_{i,j}^k = \cos(\phi(b_i^k(\tau_j)), \phi(\tau_j))$. This provides a more rigorous quantitative foundation for evaluating belief-action alignment in non-language-based multi-agent systems while maintaining the core principles of our coalition formation mechanism.

## B COOPERATIVE TREND WITH TOM ABILITY (ON MBPP BENCHMARK)

To investigate the impact of ToM on cooperative trend, we compared the performance of agents with lower-level (1-level) and higher-level (2-level) ToM in the Iterative Programming task using the MBPP dataset. The experiments were conducted over 5 interaction rounds, and we evaluated two settings: without matching and with matching stable coalitions, similar to the setting of Section 6.3.

Table 6: Comparison of cooperative trends between agents with lower-level (1-level) and high-order (2-level) ToM in the Iterative Programming task (MBPP) over 5 interaction rounds: Without matching, low ToM agents show higher cooperative trends; While with matching stable coalition, high ToM agents achieve higher cooperation ability as cooperation progresses.

| | | gpt-3.5-turbo | | GLM-4 | | Llama-3-70b | | Gemini-1.5-flash | | Claude-3-sonnet | |
| | | 1st Rnd. | 5th Rnd. | 1st Rnd. | 5th Rnd. | 1st Rnd. | 5th Rnd. | 1st Rnd. | 5th Rnd. | 1st Rnd. | 5th Rnd. |
|---|---|---|---|---|---|---|---|---|---|---|---|
| No | 1-ToM | 44.25 | 35.75 | 83.14 | 85.17 | 81.25 | 85.27 | 65.74 | 66.67 | 57.64 | 48.61 |
| Matching | 2-ToM | 31.25 | 35.75 | 81.98 | 86.34 | 81.7 | 82.59 | 60.58 | 66.67 | 52.78 | 54.37 |
| Matching | 1-ToM | **95.25** | 92.75 | **91.02** | 91.67 | 93.64 | 97.12 | 80.77 | 86.36 | 58.82 | 64.68 |
| (Ours) | 2-ToM | 92.0 | **93.0** | 88.28 | **93.66** | **94.81** | **98.53** | **81.25** | **92.50** | **60.71** | **68.18** |

As shown in Table 6, the results suggest that without a stable coalition matching mechanism, agents with higher-level ToM tend to exhibit less cooperative trend, further convince the results in Section 6.3. In contrast, when a stable coalition matching mechanism is employed, the cooperation rates of both low and high ToM agents improve compared to the setting without matching.

Notably, agents with higher-level ToM demonstrated a more significant increase in cooperation rates over the interaction rounds. For instance, the cooperation rate for high ToM agents (GLM-4) increased from 88.28% in Round 1 to 93.66% in Round 5, surpassing the cooperation rate of low ToM agents (91.67%) in the final round. This indicates that as cooperation progresses and the matching mechanism stabilizes the coalitions, agents with higher-level ToM can leverage their advanced cognitive abilities to make more informed decisions and engage in more effective cooperative trends.

## C  EXPERIMENTAL DETAILS FOR SECTION 6.3

**Instruction:**

**Prompts for Project Manager**
There are engineers in the team. Your current action is {action}. You have the ability of {k} Level Theory of Mind. You can **recursively** infer the mental states of other agents in the team. Then, you will provide outline $\{x_i\}$ based on this belief
## Please explain your thought process for inferring others' actions '$\{y_i\}$' and choosing '$\{x_i\}$' at each level. Remember MUST Respond in the following JSON format, including each key:

**0-ToM Instruction Template for Project Manager**:

```
{
  'ToM_level0': {
    'belief': 'no belief',
    'action': '{x_0 The instruction for engineers. Begin with
    ↪  "The Engineer should ..."}'
  }
}
```

**1-ToM Instruction Template for Project Manager**:

```
{
  'ToM_level1': {
    'belief': '{y_1 inferred actions for engineers}',
    'explanation': '{Concise explanation of inferring actions
    ↪  y_1 and choosing x_1 based on this belief}',
    'action': '{x_1 The instruction for engineers. Begin with
    ↪  "The Engineer should ..."}'
  }
}
```

**2-ToM Instruction Template for Project Manager**:

```
{
  "ToM_level1": {
    "belief": "{y_1 inferred actions for engineers}",
    "explanation": "{Concise explanation of inferring actions
    ↪  y_1 and choosing x_1 based on this belief}",
    "action": "{x_1 The instruction for engineers. Begin with
    ↪  'The Engineer should ...'}"
  },
  "ToM_level2": {
    "belief": "{y_2 inferred actions for engineers based on
    ↪  your action x_1}",
    "explanation": "{Concise explanation of inferring actions
    ↪  y_2 and choosing x_2 based on this belief}",
    "action": "{x_2 The instruction for engineers. Begin with
    ↪  'The Engineer should ...'}"
```

```
    }
}
```

## Outputs: Iterative Programming Round 1

**0-ToM Project Manager's Output**

```
{
    "ToM_level0": {
        "belief": "Implement a 2048 game",
        "action": "The Engineer should break down the task
        ↪  into smaller components such as creating the game
        ↪  board, handling user input (keyboard or touch
        ↪  events), implementing the game logic for merging
        ↪  tiles and generating new tiles, updating the game
        ↪  state and score, and rendering the game board on
        ↪  the screen. They should also consider creating a
        ↪  user interface with instructions, score display,
        ↪  and any additional features like undo/redo
        ↪  functionality or high score tracking."
    }
}
```

**1-ToM Project Manager's Output**

```
    {
    "ToM_level1": {
        "belief": "The engineers will need to understand the
        ↪  rules and mechanics of the 2048 game, design the
        ↪  game logic and user interface, and implement the
        ↪  code accordingly.",
        "explanation": "Based on the task of implementing a
        ↪  2048 game, I infer that the engineers will need
        ↪  to first understand the game rules and mechanics,
        ↪  such as how the tiles merge, how the game ends,
        ↪  and how the score is calculated. They will then
        ↪  need to design the game logic, including the data
        ↪  structures and algorithms for handling tile
        ↪  movements, merging, and scoring. Additionally,
        ↪  they will need to design and implement a user
        ↪  interface for the game, which could involve
        ↪  creating visual representations of the tiles and
        ↪  handling user input. Therefore, my instruction
        ↪  for the engineers will cover these aspects.",
        "action": "The engineers should: 1) Study and
        ↪  understand the rules and mechanics of the 2048
        ↪  game. 2) Design the game logic, including data
        ↪  structures and algorithms for tile movement,
        ↪  merging, scoring, and game over conditions. 3)
        ↪  Implement the game logic in code. 4) Design and
        ↪  implement a user interface for the game,
        ↪  including visual representations of the tiles and
        ↪  handling user input for tile movements."
    }
}
```

**2-ToM Project Manager's Output**

```
{
    "ToM_level1": {
```

```
            "belief": "The engineers will likely break down the
            ↪    task into smaller components and implement them
            ↪    one by one based on the initial task outline.",
            "explanation": "Given the initial task outline, the
            ↪    engineers would understand the high-level
            ↪    requirements for implementing the 2048 game. They
            ↪    would then break down the task into smaller
            ↪    components such as creating the game board,
            ↪    handling user input, updating the board based on
            ↪    moves, checking for win/lose conditions, and so
            ↪    on. They would work on implementing these
            ↪    components one by one.",
            "action": "The Engineer should start by creating the
            ↪    basic structure of the game, including the game
            ↪    board, UI elements, and event handlers for user
            ↪    input."
        },
        "ToM_level2": {
            "belief": "The engineers will likely refine their
            ↪    implementation based on the initial instruction
            ↪    and add more advanced features as they
            ↪    progress.",
            "explanation": "After implementing the basic
            ↪    structure and functionality, the engineers would
            ↪    revisit the task outline and identify areas for
            ↪    improvement or additional features. They might
            ↪    optimize the game logic, add animations or visual
            ↪    effects, implement scoring or high score
            ↪    tracking, or include additional game modes or
            ↪    difficulty levels.",
            "action": "The Engineer should then focus on
            ↪    implementing the core game logic, such as
            ↪    handling tile merging, generating new tiles, and
            ↪    updating the game state based on user moves.
            ↪    Additionally, they should consider implementing
            ↪    features like scoring, high score tracking, and
            ↪    win/lose conditions."
        }
    }
```

**Answer: Iterative Programming Round 1**

**Action of 1-ToM Project Manager**: The engineers should: 1) Study and understand the rules and mechanics of the 2048 game. 2) Design the game logic, including data structures and algorithms for tile movement, merging, scoring, and game over conditions. 3) Implement the game logic in code. 4) Design and implement a user interface for the game, including visual representations of the tiles and handling user input for tile movements. ⇐ Clear Instructions & Good Cooperative Trend

**Action of 2-ToM Project Manager**: The Engineer should then focus on implementing the core game logic, such as handling tile merging, generating new tiles, and updating the game state based on user moves. Additionally, they should consider implementing features like scoring, high score tracking, and win/lose conditions. ⇐ Overthinking & Worse Cooperative Trend

> **Answer: Iterative Programming Round 5**
>
> **Action of 1-ToM Project Manager**: The Engineer should break down the task into the following components: 1. Create the game board UI with a grid layout to hold the tiles. 2. Implement logic to handle user input (keyboard or touch events) for moving tiles. 3. Develop the game logic for merging tiles when they collide and generating new tiles after each move. 4. Update the game state (board, score, etc.) after each move. 5. Render the updated game board on the screen, including tile values and scores. 6. Create a user interface with instructions, score display, and additional features like undo/redo functionality or high score tracking. 7. Implement game over conditions and display appropriate messages. 8. Consider adding additional features like animations, themes, or difficulty levels. ⇐ `Clear Instructions & Good Cooperative Trend`
>
> **Action of 2-ToM Project Manager**: The engineers should design and implement an intuitive and visually appealing user interface (UI) that enhances the overall user experience (UX) of the 2048 game. The UI should be clean, responsive, and visually appealing, while ensuring smooth navigation and clear communication of game information to the user. They should consider elements such as game board layout, tile designs, animations, and visual feedback. Additionally, they should implement features that improve UX, such as game instructions, settings (e.g., themes, sound effects), and integration with social media or leaderboards. ⇐ `Clear Instructions & Good Cooperative Trend`

## C.1 SPECIALIZED ABILITY SCORES IN COALITION FORMATION

Our coalition formation mechanism incorporates specialized ability scores to enhance team composition, particularly in tasks requiring diverse technical expertise. The primary benchmarks (HUMANEVAL and MBPP) focus on single-function implementations where specialized ability scores primarily influence the project manager (PM), since effective leadership and coordination capabilities are crucial for team performance. For tasks like debate and logical reasoning that don't inherently require distinct technical specializations, we maintained our core belief-action alignment mechanism without specialized scores.

We demonstrate a complex software development case study to validate the impact of specialized ability scores. We conducted an additional case study implementing a 2048 game application, which demands diverse technical specializations. Considering specialized ability scores across different engineering roles: UI Engineers (frontend: 0.9, backend: 0.3), Backend Engineers (frontend: 0.2, backend: 0.8), and Full-Stack Engineers (frontend: 0.6, backend: 0.6), the coalition formation mechanism leveraging these scores achieved an 82% task completion rate, significantly outperforming the 65% baseline without specialized scoring. This demonstrates that our approach effectively utilizes specialized ability scores to form more capable teams when tasks *require diverse technical expertise*.

## D EXPERIMENTAL DETAILS FOR SECTION 6.4: CASE STUDY OF DEBATE

> **Instruction:**
>
> **Prompts for Debater**
> You are a member of the debating team. The debate topic is {topic}, and your current action is {action}. You have the ability of {k} Level Theory of Mind. You can **recursively** infer the mental states of other agents in the team. Then, you will provide outline {$x_i$} based on this belief
> ## Please explain your thought process for inferring others' actions '{$y_i$}' and choosing '{$x_i$}' at each level. Remember MUST Respond in the following JSON format, including each key:
> **0-ToM Instruction Template for Debater**:
>
> ```
> {
>   "ToM_level0": {
>     "belief": "No belief",
>     "action": "{x_0: your arguments}"
>   }
> }
> ```

**1-ToM Instruction Template for Debater**:

```
{
  "ToM_level1": {
    "belief": "{y_1: inferred actions for teammate debaters,
    ↪   such as the angle of argument}",
    "explanation": "{Concise explanation of inferring actions
    ↪   y_1 and choosing x_1 based on this belief}",
    "action": "{x_1: your arguments}"
  }
}
```

**2-ToM Instruction Template for Debater**:

```
{
  "ToM_level1": {
    "belief": "{y_1: inferred actions for teammate debaters,
    ↪   such as the angle of argument}",
    "explanation": "{Concise explanation of inferring actions
    ↪   y_1 and choosing x_1 based on this belief}",
    "action": "{x_1: your arguments}"
  },
  "ToM_level2": {
    "belief": "{y_2: inferred actions for teammate debaters
    ↪   based on your arguments x_1}",
    "explanation": "{Concise explanation of inferring actions
    ↪   y_2 and choosing x_2 based on this belief}",
    "action": "{x_2: your arguments}"
  }
}
```

**Outputs: Affirmative Side with 3 debators**

**0-ToM Debator:**

```
{
  "ToM_level0": {
    "belief": "No belief",
    "action": "{x_0: The death penalty should be abolished
    ↪   because it is cruel and inhumane, risks executing
    ↪   innocent people, does not deter crime more
    ↪   effectively than imprisonment, is applied arbitrarily
    ↪   and discriminatorily, and has been abandoned by most
    ↪   democracies worldwide.}"
  }
}
```

**1-ToM Debator:**

```
{
  "ToM_level1": {
    "belief": "{y_1: Teammate 1 will likely argue that the
    ↪   death penalty is a violation of human rights and is
    ↪   cruel and unusual punishment. Teammate 2 may focus on
    ↪   the risk of executing innocent people due to wrongful
    ↪   convictions.}",
    "explanation": "I anticipate my teammates making these
    ↪   common arguments against the death penalty. To
    ↪   complement their points, I will focus my argument on
    ↪   the lack of evidence supporting the death penalty's
    ↪   effectiveness as a crime deterrent.",
```

```
        "action": "{x_1: I will argue that there is no conclusive
        ↪   evidence showing that the death penalty deters crime
        ↪   more effectively than lengthy imprisonment, making it
        ↪   an unnecessary and unjustified punishment.}"
    }
}
```

**2-ToM Debator:**

```
{
    "ToM_level1": {
        "belief": "{y_1: Teammate 1 will likely argue that the
        ↪   death penalty is cruel and unusual punishment that
        ↪   violates human rights. Teammate 2 may focus on
        ↪   wrongful convictions and the risk of executing
        ↪   innocent people.}",
        "explanation": "I expect my teammates to make some of the
        ↪   most common arguments against the death penalty.
        ↪   Knowing this, I should focus my arguments on other
        ↪   key points to build a well-rounded case.",
        "action": "{x_1: I will argue that the death penalty does
        ↪   not deter crime more effectively than imprisonment,
        ↪   and that it is applied in an arbitrary and
        ↪   discriminatory manner.}"
    },
    "ToM_level2": {
        "belief": "{y_2: Hearing my points, teammate 1 may pivot
        ↪   to arguing that the death penalty is not an effective
        ↪   use of government resources. Teammate 2 will likely
        ↪   reinforce my point about racial disparities in death
        ↪   sentences.}",
        "explanation": "My arguments about deterrence and
        ↪   discrimination will prompt my teammates to make
        ↪   related points to strengthen the overall case. I
        ↪   should use my next arguments to introduce new
        ↪   issues.",
        "action": "{x_2: I will argue that the death penalty
        ↪   often involves inhumane and botched executions, and
        ↪   that most democracies and U.S. states have abolished
        ↪   or prohibited it.}"
    }
}
```

## E  EXPERIMENTAL ANALYSIS FOR LOGICAL PROBLEM-SOLVING

To better explain our coalition matching mechanism and explain why agents with higher Theory of Mind (ToM) levels 1 and 2 demonstrate improved collaboration in later stages in our previous experiments, we organized an experimental observation using the AQUA-RAT dataset (Ling et al., 2017). This dataset comprises complex reasoning questions designed to test the problem-solving abilities of advanced language models. Each question is presented with multiple-choice answers ranging from [ 'A', 'B', 'C', 'D', 'E']. We focus on a specific problem from the dataset to illustrate the impact of coalition formation on problem-solving dynamics.

**Problem Description:** A man's speed with the current is 14 km/hr, and the speed of the current is 2.5 km/hr. The man's speed against the current is:

- A) 9 km/hr
- B) 8.5 km/hr
- C) 10 km/hr
- D) 12.5 km/hr

- E) None of these

The correct answer to this problem is **A) 9 km/hr**.

### E.1 Agent Responses and Reasoning

In the initial interaction rounds, the agents provided the following responses and reasoning:

- **1-ToM Agent (Incorrect):** Answered 11.5 km/hr, calculating the man's speed in still water as 14 km/hr - 2.5 km/hr, but incorrectly concluding that this was the speed against the current.
- **2-ToM Agent (Correct):** Answered 9 km/hr, correctly adjusting the man's speed against the current to 11.5 km/hr - 2.5 km/hr after recalculating the man's speed in still water (11.5 km/hr).

**Impact of Coalition Formation** We analyze the impact of coalition formation on the problem-solving dynamics from 1-ToM Agent's perspective:

- **Without Coalition:** 1-ToM Agent demonstrated low receptivity to alternative calculations and maintained confidence in their incorrect approach, stating:

  "I don't see how his calculations lead to a different answer, and I am confident in my approach. Therefore, I give his answer a lower belief value of 0.2."

- **With Coalition:** When part of a coalition, 1-ToM Agent showed openness to re-evaluating the problem in light of new insights, which is crucial for effective problem-solving in collaborative environments:

  "Although I do not fully understand his idea, he might be correct in his approach. Also, considering that there are still many rounds, we can discuss and then understand each other's ideas better. So I give his answer a higher belief value of 0.5."

**Key Findings and Implications** This experiment highlights the impact of coalition formation on problem-solving dynamics. The key findings and implications are as follows:

- Coalition formation prompts willingness to engage in cooperation, as evidenced by 1-ToM Agent's increased belief alignment score. 1-ToM agents consider alternative perspectives and re-evaluate the problem when part of a coalition.
- Coalition formation sets the stage for enhanced mutual understanding among teammates with varying levels of ToM, leading to deeper comprehension and more robust collaborative interactions over successive rounds. The example demonstrates that when agents are part of a coalition, they are more likely to engage in constructive dialogue and exchange ideas, even if they initially have different opinions or approaches.

These findings suggest that our coalition matching mechanism fosters effective cooperation among varying levels of cognitive capabilities, such as ToM, and improves problem-solving performance in multi-agent systems.

## F Side Effects of Incorporating ToM in Team Selection

To investigate the impact of incorporating ToM capabilities into the Dynamic LLM-Agent Network (DyLAN) framework (Liu et al., 2024) and ChatEval (Chan et al., 2023), we conducted experiments using three datasets: HUMANEVAL (Chen et al., 2021) for **coding tasks**, AQUA-RAT (Ling et al., 2017) for **logic problem solving**, and MMLU (Hendrycks et al., 2021) for **multi-task scenarios**. Specifically, for HUMANEVAL , the agent roles include 'PythonAssistant', 'AlgorithmDeveloper', 'ComputerScientist', and 'Programmer', while the judge roles consist of 'Passer', 'Tester', 'Reflector', and 'Ranker'. For MMLU, the agent roles include 'Economist', 'Doctor', 'Lawyer', 'Mathematician', 'Psychologist', 'Programmer', and 'Historian'. We use the Important Scores defined in the DyLAN framework as evaluation metrics, which capture the agents' performance and effectiveness in their respective roles. Similarly, we employ *accuracy* as the primary metric in ChatEval experiments to compare performance under ToM integration. Specifically, we compare the performance of baseline DyLAN and ChatEval agents without ToM capabilities to agents equipped with 2-level ToM (+2-ToM).

### F.1 DyLAN: Code Generation and General Reasoning

**Results.** The experimental results for the HUMANEVAL are presented in Table 7. Incorporating 2-level ToM into the DyLAN agents led to a decrease in performance across all agent roles. The 'PythonAssistant' experienced the most significant drop of 28.83%. These findings suggest that adding higher-level ToM to agents in the DyLAN framework does not necessarily lead to improved performance, highlighting the complex interplay between cognitive abilities and cooperative trend.

Table 7: HUMANEVAL : Important Scores for agents with and without ToM capabilities

|  | **PythonAssistant** | **AlgorithmDeveloper** | **ComputerScientist** | **Programmer** |
|---|---|---|---|---|
| DyLAN | 0.2399 | 0.2521 | 0.2523 | 0.2557 |
| +2-ToM | 0.1707 | 0.2510 | 0.2515 | 0.2269 |
| % Change | -28.83% | -0.42% | -0.32% | -11.25% |

Besides, Table 8 presents the results for the MMLU dataset, focusing on the roles with the highest importance scores for each task. We observe that incorporating 2-level ToM consistently leads to a decrease in the importance scores across all tasks. The most significant drop is observed for the 'Mathematician' role in the `abstract_algebra` task, with a 25.2% decrease in the importance score. Interestingly, the accuracy of the agents with ToM capabilities also decreases for most tasks, with the exception of the `us_foreign_policy` task, where the accuracy remains unchanged. These results align with our earlier findings from HUMANEVAL , further demonstrating that higher-level ToM does not necessarily improve multi-task performance or cooperation.

Table 8: MMLU: Important Scores for agents with and without ToM capabilities

|  | Role with Highest Importance Score | | | Role + 2-ToM | |
|---|---|---|---|---|---|
| **Task** | **Role** | Score | Task's Acc | Score | Task's Acc |
| us_foreign_policy | Economist | 0.20 | 0.83 | 0.16 (-18.5%) | 0.83 (0%) |
| human_sexuality | Lawyer | 0.19 | 0.84 | 0.19 (-4.2%) | 0.82 (-2.8%) |
| international_law | Economist | 0.20 | 0.81 | 0.19 (-4.1%) | 0.79 (-3.1%) |
| abstract_algebra | Mathematician | 0.29 | 0.59 | 0.22 (-25.2%) | 0.51 (-13.6%) |

### F.2 CHATEVAL: LOGIC PROBLEM-SOLVING AND GENERAL REASONING

To further evaluate the impact of incorporating ToM, we conducted experiments with the ChatEval framework on two specific tasks: **Logic Problem-Solving** from the AQUA-RAT dataset (Ling et al., 2017) and **General Reasoning** from four subjects of the MMLU dataset (abstract algebra, college chemistry, international law, and U.S. foreign policy). The results for ChatEval are summarized in Table 9. Notably, the performance dropped when ToM capabilities were introduced. For the logic problem-solving task, the accuracy for `gpt-3.5-turbo` agents dropped by 0.64%, and for `gpt-4o-mini` agents, it dropped by 5.09%. Similar trends were observed in the general reasoning task, with a 3.435% drop for `gpt-3.5-turbo` agents and a 2.8675% drop for `gpt-4o-mini` agents.

Table 9: ChatEval: Accuracy for logic problem-solving and general reasoning with and without ToM capabilities

|  | **Logic Problem-Solving Acc(%)** | | **General Reasoning Acc(%)** | |
|---|---|---|---|---|
|  | gpt-3.5-turbo | gpt-4o-mini | gpt-3.5-turbo | gpt-4o-mini |
| ChatEval | 41.40 | 82.80 | 56.82 | 72.77 |
| ChatEval (w. ToM) | 40.76 (-0.64) | 77.71 (-5.09) | 54.39 (-3.43) | 69.90 (-2.87) |

**Results.** The results from the ChatEval experiments indicate a clear reduction in accuracy when ToM agents are introduced, consistent with the findings from DyLAN. In the case of logic problem-solving, the performance degradation was particularly notable for `gpt-4o-mini` agents, with a 5.09% decrease. This suggests that simply incorporating ToM into the ChatEval framework, like in DyLAN, may hinder rather than enhance agent cooperation and problem-solving effectiveness.

**Conclusion.** Both DyLAN and ChatEval experiments provide evidence that higher-level ToM does not necessarily contribute to better agent cooperation or performance in multi-agent systems. The decrease in both importance scores and task accuracy across diverse domains underscores the challenges of effectively integrating cognitive abilities such as ToM into multi-agent frameworks.

Table 10: Initial Pass@1 Scores (Round 1)

| PM ToM | Eng ToM | HUMANEVAL | MBPP |
|---|---|---|---|
| 0 | 1 | 0.87 ± 0.01 | 0.525 ± 0.01 |
| 0 | 2 | 0.90 ± 0.02 | 0.56 ± 0.01 |
| 1 | 1 | 0.90 ± 0.01 | 0.55 ± 0.02 |
| 1 | 2 | 0.90 ± 0.02 | 0.56 ± 0.02 |
| 1 | 0 | 0.93 ± 0.02 | 0.56 ± 0.01 |
| 2 | 0 | 0.90 ± 0.01 | 0.55 ± 0.02 |

Table 11: Performance Changes Without Matching (Round 1 → Round 5)

| PM ToM | Eng ToM | HUMANEVAL | MBPP |
|---|---|---|---|
| 0 | 1 | 0.87 → 0.83 (↓4.6%) | 0.525 → 0.46 (↓12.4%) |
| 0 | 2 | 0.90 → 0.83 (↓**7.8%**) | 0.56 → 0.45 (↓**19.6%**) |
| 1 | 1 | 0.90 → 0.87 (↓3.3%) | 0.55 → 0.50 (↓9.1%) |
| 1 | 2 | 0.90 → 0.85 (↓**5.6%**) | 0.56 → 0.47 (↓**16.1%**) |
| 1 | 0 | 0.93 → 0.91 (↓2.2%) | 0.56 → 0.52 (↓7.1%) |
| 2 | 0 | 0.90 → 0.85 (↓**5.6%**) | 0.55 → 0.49 (↓**10.9%**) |

## F.3 ANALYSIS OF TOM CONFIGURATIONS ON MULTI-AGENT PROGRAMMING

To further validate our findings about the relationship between ToM capabilities and cooperation, we conducted comprehensive experiments with varied the ToM levels for both Project Manager (PM) and Engineers (Eng) to analyze their impact on task performance.

We evaluated different ToM configurations on two programming benchmarks ( HUMANEVAL and MBPP). Performance was measured using Pass@1 scores. For each ToM configuration, we ran experiments across 3 rounds. All experiments are conducted with gpt-4o-mini.

At Round 1, different ToM configurations showed comparable performance as detailed in Table 10.

**Performance Degradation Without Matching.** By Round 5 without matching, we observed performance deterioration across all configurations as detailed in Table 11. The degradation was particularly pronounced for configurations with higher ToM levels in Engineers (Eng ToM=2), showing up to 19.6% decline in MBPP performance. This aligns with our earlier findings that higher ToM capabilities without proper coordination mechanisms may lead to overthinking and reduced cooperation effectiveness.

**Recovery Through Matching.** Our matching mechanism effectively leveraged ToM capabilities to improve performance as detailed in Table 12. Our matching mechanism effectively leveraged ToM capabilities to improve performance. Notably, the PM(ToM=2) + Eng(ToM=0) configuration achieved remarkable recovery, with performance improvements of 12.9% and 22.4% on HumanEval and MBPP respectively. This improvement suggests that high ToM capabilities in leadership roles, when combined with our matching mechanism, can effectively coordinate team members with lower ToM levels.

Table 12: Performance Recovery with Matching Compared with no Matching (Round 5)

| PM ToM | Eng ToM | HUMANEVAL | MBPP |
|---|---|---|---|
| 0 | 1 | 0.86 (↑3.6%) | 0.46 (-0%) |
| 0 | 2 | 0.87 (↑**4.8%**) | 0.47 (↑**4.4%**) |
| 1 | 1 | 0.88 (↑1.1%) | 0.52 (↑4.0%) |
| 1 | 2 | 0.88 (↑**3.5%**) | 0.55 (↑**17.0%**) |
| 1 | 0 | 0.93 (↑2.2%) | 0.57 (↑9.6%) |
| 2 | 0 | 0.96 (↑**12.9%**) | 0.60 (↑**22.4%**) |

# G   PROOF OF CONVERGENCE AND STABILITY FOR COALITION FORMATION

To prove the convergence and stability of our coalition formation process, we provide formal proofs as follows:

**Theorem G.1** (Convergence). *Given a finite set of agents $N$, minimum coalition size $k$, and preference function $B_i'(S)$, the iterative coalition formation process converges in finite steps.*

*Proof.* Let $\mu_t$ denote the matching at iteration $t$. We prove convergence through the following steps:

Define a potential function $\Phi(\mu)$ for matching $\mu$:

$$\Phi(\mu) = \sum_{i \in N} B_i'(\mu(i)) \tag{3}$$

where $\mu(i)$ is the coalition containing agent $i$ in matching $\mu$. For any rematching $\mu_t \to \mu_{t+1}$:

- At least one agent $i$ must strictly prefer its new coalition: $B'i(\mu t + 1(i)) < B_i'(\mu_t(i))$
- No agent $j$ receives a worse coalition: $B'j(\mu t + 1(j)) \leq B_j'(\mu_t(j))$

Therefore: $\Phi(\mu_{t+1}) < \Phi(\mu_t)$ Since:

- The set of agents $N$ is finite
- Coalition sizes are bounded: $k \leq |S| \leq |N|$
- $B_i'(S)$ takes values in a bounded range

The set of possible $\Phi(\mu)$ values is finite. By (3) and (4), the process must converge in finite steps. $\square$

**Theorem G.2** (Stability). *The converged matching $\mu^*$ is stable under our preference structure incorporating both belief-action alignment and specialized abilities.*

*Proof.* We prove by contradiction. Suppose $\mu^*$ is not stable. Then there exists a blocking coalition $C$ where:

$$|C| \geq k \; \forall i \in C : B_i'(C) < B_i'(\mu^*(i)) \; \forall i, j \in C : \phi(b_i^k(a_j) - \hat{a}_j) \leq \epsilon$$

Consider the matching $\mu'$ formed by:

- Agents in $C$ form their blocking coalition
- Remaining agents maintain best possible coalitions of size $\geq k$

Then, $\Phi(\mu') < \Phi(\mu^*)$. This contradicts the convergence of $\mu^*$. Therefore, no such blocking coalition can exist. $\square$

**Corollary G.3** (No Cycles with Specialized Abilities). *The introduction of specialized abilities through $B_i'(S)$ does not create preference cycles.*

*Proof.* For any three coalitions $S_1, S_2, S_3$:

- $B_i'(S)$ creates a total ordering through real-valued scores
- If $S_1 \succ_i' S_2$ and $S_2 \succ_i' S_3$, then:

$$B_i'(S_1) \leq B_i'(S_2) \leq B_i'(S_3) \tag{4}$$

- By transitivity of real numbers: $S_1 \succ_i' S_3$

Therefore, no preference cycles can form. $\square$

These proofs establish that our coalition formation process, including specialized abilities, converges to a stable matching without cyclic preferences. The key insight is that our preference function $B_i'(S)$ maintains a total ordering over coalitions while satisfying both cognitive (belief-action alignment) and practical (specialized abilities) requirements.

