# OpenReview forum: "Cognitive Insights and Stable Coalition Matching for Fostering Multi-Agent Cooperation"
_ICLR.cc/2025/Conference — Submitted to ICLR 2025_

### Official Review · Reviewer_rCxA · 2024-11-02

**Soundness:** 2
**Presentation:** 2
**Contribution:** 2
**Rating:** 3
**Confidence:** 4

**Summary:**

This paper studies the concept of theory of mind (ToM) in the context of one LLM agent forming and managing coalitions of other LLM agents. They show that prompting the first LLM agent to engage in 2-level reasoning about the others' beliefs can actually _hinder_ performance compared to 1-level reasoning (cf. the general concept of k-level reasoning). They introduce a method of comparing and matching agents based on their ability to predict each other's actions. Their experiments study how the use of that metric in forming coalitions of LLM agents impacts the agents' ability to solve problems in the domains of programming, logic, and general reasoning.

**Strengths:**

The questions underlying this paper are certainly interesting. While much work has focused on the improved cooperation using ToM, few (to my knowledge) have investigated how this might hinder cooperation, at least not (again, to my knowledge) in the context of LLM agents. More generally, the issue of intelligent coalition formation in this context is an interesting problem and one that I believe will have increasing real-world relevance in the coming years. The idea of alignment of beliefs in order to solve this problem is natural and original (again, that is to the best of my knowledge, though I also would not be at all surprised if a version of this had been studied before in the game theory or multi-agent systems literatures, outside the context of LLMs). I also appreciated the effort the authors put in to studying a relatively wide variety of tasks, models, and frameworks for multi-LLM problem-solving. The presentation of their results was largely clear.

**Weaknesses:**

Unfortunately, despite its positive aspects, I think the paper does have some significant issues. In what follows I have attempted to cluster these and order them approximately by importance.

**Matching Algorithm Confusions**

The matching algorithm seems underspecified in several places, and it is not always clear what the authors are actually doing. Moreover, I didn't feel that the underlying theoretical principles were always appropriate. More concretely:
- The authors start by talking about the set of an agent $i$'s partners $\mu(i)$, but based on their matching algorithm they seem to implicitly assume that $\mu(i)$ is always a singleton (see Equation 2). Otherwise, matchings are stable only when player $i$ would not prefer to be partnered with $j$ over its current _set_ of partners $\mu(i)$. But what would that mean? Are preferences between sets of agents instead of single agents? If so, the relevant comparison would presumably be that agent player $i$ would not prefer to be partnered with any _set_ of other agents over its current _set_ of partners $\mu(i)$.
- In line 9 of algorithm 1, what happens after agents signal a desire to re-match? What if the belief misalignment measure is greater than the tolerance $\epsilon$ for all other agents? Does the agent end up in a singleton coalition? The authors state that the iterative process of coalition formation ends in a stable matching but they do not actually prove this. Especially with the introduction of preferences based on differing skills (see the next point), I actually suspect that it would be trivial to create a cyclic matching problem.
- The preference order described in equation 4 are based on agents having different skill levels $\alpha_i$ on different tasks, but where do these skill levels come from? More importantly, if the point is to match agents with complementary skills, why does the matching algorithm only compare agents' skills on a _single_ task?
- Minor: the authors say on line 244 that the alignment between beliefs and actions is not mathematical subtraction, despite them denoting it that way. I would strongly suggest not denoting it using subtraction to begin with and being more explicit about what the distance measure here actually is.

**Strength of Motivating Claim**

The authors' motivating claim is that lower-level ToM abilities may improve the ability of agents to cooperate beyond higher-level ToM abilities. Their justification for this is a setting where one agent -- a "Project Manager" (PM) -- is instructed to use either 1-level or 2-level reasoning to organise several other agents (all of which are instructed use 0-level reasoning). But this essentially means that in the latter case the PM is being instructed to reason about the agents acting in a way that they do not in fact act. Explained this way, it is still somewhat interesting but by no means surprising that the PM is less successful when prompted to reason using higher-level ToM. Essentially, $k$-level reasoners are designed to best respond to $(k-1)$-level reasoners, not $(k-2)$-level reasoners.
- Relatedly, looking through the actual LLM outputs included in the appendices, the level-2 ToM responses seem quite strange. They are worded as if they are predicated on the other agents _actually observing_ actions in advance, rather than _anticipating_ instructions. I am not really sure what is going on here, but reading through it was not at all surprising that the higher ToM agents performed less well on the task, as they appeared to be being mis-instructed.
- As a final sub-point on this topic, I suggest that the authors also benchmark against a 0-level PM and against settings where the agents are 1-level or 2-level reasoners, at least for their motivating experiments described in Table 1.

**Missing Experimental Details**

There are several (relatively minor) aspects missing from the discussion and presentation of the experiments that, if present, would improve the paper.

- There are no error bars or reports of standard errors for the experimental results, making it difficult to interpret their statistical significance.
- I assume the ToM level for debating agents arguing for the negative side is 0, but it would be good to clarify this.
- Once coalitions are formed, how do the prompts/instructions given to the agents in different coalitions actually change?
- How many agents are actually present in the various settings, and what are the sizes of the coalitions that are formed?

**Game-Theoretic Reasoning and Precision of Claims**

This is a relatively minor, but a few times I found myself slightly frustrated by the authors claims, which I believe did not fully take into account the relevant game-theoretic concepts (see also the confusing use of what appears to be a binary matching algorithm for n-player coalition formation, described further above).

- A key example is the authors' claim that the so-called "Fraction of Trust Members (FTM)" is a good measure of what they term the "cooperative trend"  (N.B. to be grammatical, this should probably be "Trusted" not "Trust", though it is not actually clear what the relevance of the concept of "trust" even is here). But belief alignment by itself does not imply higher levels of cooperation. I may have perfectly accurate beliefs about what you are going to do in a two-player zero-sum game (where cooperation is definitionally impossible). Thus, it is clearly not true that in general "a higher FTM value [indicates] a more cooperative agent", as claimed in line 410/411.
- Relatedly, the authors talk about ToM improving cooperation but is more about ToM improving the facilitation/management skills of a single PM agent. This is also a very interesting and valid topic of study, but I suggest the authors change the phrasing slightly throughout the paper to better reflect the rather narrow form of cooperation problem they consider. Indeed, my understanding is that the authors largely focus on the case where only one agent (a PM) is imbued with ToM.

**Questions:**

Please see the Weaknesses section for my questions. I also welcome the authors to correct any misunderstandings I may have about their paper.

---

> ### Author Response · Authors · 2024-11-21
> **Re. A. Matching Algorithm Confusions**
>
> > A-W1. The authors start by talking about the set of an agent $i$'s partners $\mu(i)$, but based on their matching algorithm they seem to implicitly assume that $\mu(i)$ is always a singleton (see Equation 2). Otherwise, matchings are stable only when player $i$ would not prefer to be partnered with $j$ over its current *set* of partners $\mu(i)$. But what would that mean? Are preferences between sets of agents instead of single agents? If so, the relevant comparison would presumably be that agent player $i$ would not prefer to be partnered with any *set* of other agents over its current *set* of partners $\mu(i)$.
>
> **R.A-W1**. Matching Implementation:
>         We sincerely think the reviewer’s thoughtful feedback on the formulation. In our original manuscript, we focus on a one-to-many matching structure, specifically motivated by the Project Manager (PM) and Engineers scenario: **PM Role (One Side,** Single PM with ToM capabilities) and **Engineer Role (Many Side).** In our implementation, we enforce a minimum coalition size to ensure effective cooperation. We maintain $|\mu(i)| \geq k$, where k is a predefined #min_coalition_size -1 (typically #min_coalition_size=$\lceil N/2 \rceil$ in our setting, where N is the total number of agents in our experiments).
>
> **Generalization to Other Settings**. We acknowledge that our implementation also considers other scenarios (like debates where **Equal-role agents are forming teams**). To accommodate **both hierarchical and peer-based scenarios**, we propose a generalized formulation adapted to our implementation:
>
> - **Preference Structure:** For agent $i$, preferences over coalitions are defined by belief-action alignment scores:
>         $B\_i(S) = \frac{1}{|S|} \sum\_{j \in S} \phi(b^k\_i(a\_j) - \hat{a}\_j)$
>         For two potential coalitions $S$$_1, S_2$:
>         $S\_1 \succ\_i S\_2 \Leftrightarrow B\_i(S\_1) < B\_i(S\_2)$
> - **Stability**: A matching $\mu$ is stable if there exists no blocking coalition $C \subseteq N$ where:
>             (1). $|C| > k$ (minimum size requirement)
>             (2). $\forall i \in C: C \succ\_i \mu(i)$ (coalition preferred by all members)
> - **Generalization**: we discuss the specialized scores in PM and Eng setting in Appendix C.1, where specialized ability scores primarily influence the PM, since effective leadership and coordination capabilities are crucial (as also evidenced by extensive evaluation in Appendix F.3, which detailed in our **response R.B-W2.**)
>
> We updated our manuscript according to the reviewer's constructive feedback, detailed on pages 5 & 6, highlighted in purple color.
>
> > A-W2. The preference order described in equation 4 are based on agents having different skill levels $\alpha_i$ on different tasks, but where do these skill levels come from? More importantly, if the point is to match agents with complementary skills, why does the matching algorithm only compare agents' skills on a *single* task?
> >
> We will clarify the reviewer's concert point by point:
>
> 1. Source of Skill Levels ($\alpha_j$):
> The skill levels come from agents' self-assessment based on their assigned roles and capabilities. For example, in the programming task, an Engineer with testing expertise = 0.9 for testing tasks.
> 2. Our current work focuses on one **task-specific requirement** in each (cooperation) round. For instance: If the **current agent i needs testing expertise**:
>     - $\alpha_j$_testing (0.9) > $αₘ$_testing (0.5)
>     - We will extend our work by incorporating multi-task skill vectors in the future.

---

> > ### Author Response · Authors · 2024-11-21
> > **Re. A-W3. Matching Algorithm Confusions**
> >
> > > A-W3. In line 9 of algorithm 1, what happens after agents signal a desire to re-match? What if the belief misalignment measure is greater than the tolerance $\epsilon$ for all other agents? Does the agent end up in a singleton coalition? The authors state that the iterative process of coalition formation ends in a stable matching but they do not actually prove this. Especially with the introduction of preferences based on differing skills (see the next point), I actually suspect that it would be trivial to create a cyclic matching problem.
> > >
> >
> > **R.A-W3.** We thank the reviewer’s detailed comments. We will address each question one by one.
> >
> > > In line 9 of algorithm 1, what happens after agents signal a desire to re-match?
> > When agents signal a desire to re-match, our algorithm:
> > >
> > - Triggers a new cooperation round and updates the preference orders based on the latest belief-action alignments
> > - If a stable matching is found, proceed with the new coalitions
> > - If no stable matching is found, keep the current coalitions for one more round
> >
> > > What if the belief misalignment measure is greater than the tolerance $\epsilon$ for all other agents? Does the agent end up in a singleton coalition?
> > >
> >
> > **For handling Universal Misalignment**:
> >
> > - The primary goal to use $\varepsilon$ is to **reduce the search space**
> >
> > When belief misalignment exceeds $\varepsilon$ for all potential partners, our implementation:
> >
> > - The preference order can still established only by relying on the alignment score.
> > - We also use #min_coalition_size (default is $\lceil N/2 \rceil$) so that at least #min_coalition_size agents are in the coalition. This is the higher priority than “reducing the search space”. The tolerance then will be adjusted.
> >
> > **Comprehensive Example.** We use the following example to illustrate:
> >
> > - Consider 6 agents: {A, B, C, D, E, F}, #min_coalition_size=2, Initial tolerance $ε$ = 0.3, and Belief-Action Alignment Scores as follows:
> >
> >     ```
> >     A's scores:
> >     A -> B: 0.1  (good alignment)
> >     A -> C: 0.2  (good alignment)
> >     A -> D: 0.25 (good alignment)
> >     A -> E: 0.4  (poor alignment)
> >     A -> F: 0.45 (poor alignment)
> >
> >     B's scores:
> >     B -> C: 0.15 (good alignment)
> >     B -> D: 0.35 (poor alignment)
> >     B -> E: 0.1  (good alignment)
> >     B -> F: 0.2  (good alignment)
> >     ```
> >
> > - Case 1: Using ε to Reduce Search Space
> >     1. For agent A:
> >         - Only consider partners with scores ≤ ε (0.3)
> >         - Candidate pool: {B, C, D}
> >         - Reduces from 10 possible 3-agent combinations to only 1: {A,B,C}
> >     2. For agent B:
> >         - Candidate pool based on ε: {C, E, F}
> >         - Significantly reduces coalition possibilities
> > - Case 2: Without Using ε (Just Sorting)
> >     - Must evaluate all possible 3-agent combinations
> >     - For just agent A: Need to check all combinations:
> >     {A,B,C}, {A,B,D}, {A,B,E}, {A,B,F}, {A,C,D}, {A,C,E}, etc.
> > - **Key Insight**
> >     - With ε: O(k) comparisons where k is number of agents within tolerance
> >     - Without ε: O(n choose #min_coalition_size) comparisons where n is total number of agents
> >
> > > The authors state that the iterative process of coalition formation ends in a stable matching but they do not actually prove this. Especially with the introduction of preferences based on differing skills (see the next point), I actually suspect that it would be trivial to create a cyclic matching problem.
> > >
> >
> > We appreciate the reviewer's concern about potential cyclic preferences in our coalition formation mechanism.
> >
> > For the revised generalized formulation of **many-to-many matching (page 5 & 6),** let us provide a rigorous proof of stability and acyclicity:
> >
> > - **Key Properties**
> >     1. **Property 1 (Well-Defined Hybrid Scoring):** For any coalition S and agent i:
> >
> >         $B'_i(S) = B_i(S) + \lambda \cdot \alpha(S)$
> >
> >         where:
> >
> >         - $B_i(S)$: belief-action alignment (continuous)
> >         - $\alpha(S)$: specialized ability score (bounded)
> >         - $\lambda$: scaling factor
> >     2. **Property 2 (Strict Preference Ordering):** For coalitions S₁, S₂: $S_1 \succ_i' S_2 \iff B'_i(S_1) < B'_i(S_2)$
> > - **Proof of Acyclicity**
> >
> >     **Theorem 1:** The preference structure cannot create cycles.
> >
> >     *Proof by Contradiction:*
> >
> >     1. Assume cycle exists: $S₁ \succ_i' S₂ \succ_i' S₃ \succ_i' S₁$
> >     2. By definition:
> >         - $B'ᵢ(S₁) < B'ᵢ(S₂)$
> >         - $B'ᵢ(S₂) < B'ᵢ(S₃)$
> >         - $B'ᵢ(S₃) < B'ᵢ(S₁)$
> >     3. Transitivity of real numbers implies:
> >     $B'ᵢ(S₁) < B'ᵢ(S₁)$ → Contradiction
> > - **Convergence to Stable Matching**
> >
> >     **Theorem 2:** The iterative process converges to a stable matching.
> >
> >     The stable matching is guaranteed by:
> >
> >     - Finite improvement path
> >     - Strict preference ordering
> >     - Size constraint k prevents degenerate solutions
> >
> > According to the reviewer's feedback, we provided the completed proof in Appendix G of the updated manuscript.

---

> ### Author Response · Authors · 2024-11-21
> **Re. B-W1. Strength of Motivating Claim**
>
> > B-W1. Relatedly, looking through the actual LLM outputs included in the appendices, the level-2 ToM responses seem quite strange. They are worded as if they are predicated on the other agents *actually observing* actions in advance, rather than *anticipating* instructions. I am not really sure what is going on here, but reading through it was not at all surprising that the higher ToM agents performed less well on the task, as they appeared to be being mis-instructed.
> >
>
> **R.B-W1.**
> We sincerely thank the reviewer for these insightful technical points about ToM levels and agent interactions. We'll address each point:
>
> - **ToM implementation** works recursively over multiple interactions. When we say "observing" in level-2 ToM responses, this refers to observing outcomes from **previous interaction** **rounds** to inform **predictions about future actions of other agents**. This creates a recursive reasoning chain:
>     - Round 1: Initial beliefs
>     - Round 2: Update beliefs based on Round 1 observations
>     - Round 3: Further updates incorporating both previous rounds
>     And so on...
> - To make the exact format of beliefs and nested reasoning processes explicit, we expand our formulation to show the recursive structure for an LLM agent $i$ at cooperation round $R$ (This revision is updated in our manuscript page 4 & 5).
> We revise its $k$-level ToM function (Eq. 1) as:
>
>     $\text{ToM}\_i^k(o\_i^{1:R}, \hat{a}\_{-i}^{1:R-1}, \\{b^{k-1}\_{i, R}(a\_m^R)\\}\_{m \neq i}) := b\_{i, R}^k$
>
>     where:
>
>     - $o_i^{1:R}$ represents agent $i$'s observation history up to round $R$, including current task state, self-actions, and collaborative teammates.
>     - $\hat{a}_{-i}^{1:R-1}$ represents other agents' action history up to round $R-1$
>     - $\\{b^{k-1}\_{i, R}(a\_m^R)\\}\_\{m \neq i\}$ captures agent $i$'s prediction of agent $m$'s action at round $R$ based on $(k-1)$-level ToM reasoning:
>     $b^{k-1}\_{i, R}(a\_m^R) = p(a\_m^R | \text{ToM}\_i^{k-1}(o\_i^{1:R}, \hat{a}\_{-i}^{1:R-1}, \\{b^{k-2}\_{i, R}(a\_l^R)\\}\_{l \neq i}))$
>
> - Moreover, we want to emphasize that our core contribution is the coalition formation mechanism as a plug-and-play approach for improving multi-agent cooperation, rather than establishing fundamental properties of ToM levels.

---

> > ### Author Response · Authors · 2024-11-21
> > **Re. B-W2. Strength of Motivating Claim**
> >
> > > B-W2. As a final sub-point on this topic, I suggest that the authors also benchmark against a 0-level PM and against settings where the agents are 1-level or 2-level reasoners, at least for their motivating experiments described in Table 1.
> > >
> >
> > **R.B-W2.**
> > We sincerely thank the reviewer for the thoughtful feedback. We agree that evaluating different ToM configurations is crucial for examining ToM’s impacts on cooperation:
> >
> > We have added comprehensive experiments incorporating **different ToM configurations** for Project Manager (PM) and Engineers (Eng) and track **performance metrics** (Pass@1) to validate cooperation effects (detailed in Appendix F.3 of the updated manuscript).
> >
> > The following provides a summary of our new evaluation results (key results are presented in Table 2 & 3):
> >
> > ### 1. Initial Performance Comparison
> >
> > **Table 1: Initial Pass@1 Scores (Round 1) on HumanEval and MBPP**
> >
> > | PM ToM | Eng ToM | HumanEval | MBPP |
> > | --- | --- | --- | --- |
> > | 0 | 1 | 0.87 ± 0.01 | 0.525 ± 0.01 |
> > | 0 | 2 | 0.90 ± 0.02 | 0.56 ± 0.01 |
> > | 1 | 1 | 0.90 ± 0.01 | 0.55 ± 0.02 |
> > | 1 | 2 | 0.90 ± 0.02 | 0.56 ± 0.02 |
> > | 1 | 0 | 0.93 ± 0.02 | 0.56 ± 0.01 |
> > | 2 | 0 | 0.90 ± 0.01 | 0.55 ± 0.02 |
> >
> > **Key Observation**: Similar initial performance across ToM configurations.
> >
> > ### 2. Performance Degradation Without Matching
> >
> > **Table 2: Pass@1 Score Changes Without Matching (Round 1 → Round 5)**
> >
> > | PM ToM | Eng ToM | HumanEval Change | MBPP Change |
> > | --- | --- | --- | --- |
> > | 0 | 1 | 0.87 → 0.83 (↓4.6%) | 0.525 → 0.46 (↓12.4%) |
> > | 0 | 2 | **0.90 → 0.83 (↓7.8%)** | **0.56 → 0.45 (↓19.6%)** |
> > | 1 | 1 | 0.90 → 0.87 (↓3.3%) | 0.55 → 0.50 (↓9.1%) |
> > | 1 | 2 | **0.90 → 0.85 (↓5.6%)** | **0.56 → 0.47 (↓16.1%)** |
> > | 1 | 0 | 0.93 → 0.91 (↓2.2%) | 0.56 → 0.52 (↓7.1%) |
> > | 2 | 0 | **0.90 → 0.85 (↓5.6%)** | **0.55 → 0.49 (↓10.9%)** |
> >
> > **Key Finding**: Higher ToM configurations show larger performance drops without matching, supporting our claim that raw ToM capabilities may actually hinder sustained performance.
> >
> > ### 3. Recovery with Matching Mechanism
> >
> > **Table 3: Performance Recovery with Matching (Round 5)**
> >
> > | PM ToM | Eng ToM | HumanEval | MBPP |
> > | --- | --- | --- | --- |
> > | 0 | 1 | 0.86 (↑3.6%) | 0.46 (±0%) |
> > | 0 | 2 | **0.87 (↑4.8%)** | **0.47 (↑4.4%)** |
> > | 1 | 1 | 0.88 (↑1.1%) | 0.52 (↑4.0%) |
> > | 1 | 2 | **0.88 (↑3.5%)** | **0.55 (↑17.0%)** |
> > | 1 | 0 | 0.93 (↑2.2%) | 0.57 (↑9.6%) |
> > | 2 | 0 | **0.96 (↑12.9%)** | **0.60 (↑22.4%)** |
> >
> > **Key Result**: Our matching mechanism effectively leverages ToM capabilities, **PM(ToM=2) + Eng(ToM=0) with matching achieves best sustained performance.**

---

> > > ### Author Response · Authors · 2024-11-21
> > > **Re. C. Missing Experimental Details**
> > >
> > > We thank the reviewer's comments, which have helped us clarify several important aspects of our experimental settings.
> > >
> > > > C-W1. I assume the ToM level for debating agents arguing for the negative side is 0, but it would be good to clarify this.
> > > >
> > >
> > > **R.C-W1.**
> > >
> > > Yes. As stated in our original manuscript “affirmative team with ToM against a team without ToM”. Thus, ToM level for the negative side is 0 in our evaluation.
> > >
> > > Additionally, we conducted symmetric experiments by placing ToM agents on the negative side while keeping the affirmative side without ToM capabilities.
> > >
> > > - As shown in the table, the performance of our method with ToM and Matching showed a win rate of **36.36%.**
> > > - This result mirrors our previous findings where ToM agents were on the affirmative side, demonstrating that the effectiveness of our coalition formation mechanism is consistent regardless of debate sides.
> > >
> > > | Setting | Win Rate |
> > > | --- | --- |
> > > | No-ToM | 34.55% (65.45% Win Rate for the affirmative side) |
> > > | ToM w.o. Matching | 25.45% |
> > > | ToM w. Matching (Ours) | **36.36%** |
> > >
> > > > Minor: The acronym FTM (Fraction of trust members/Frequency of team matching) is used multiple times making some sections difficult to understand.
> > > >
> > >
> > > We have update the manuscript using *Fraction of trusted members* only.
> > >
> > > > C-W2. Once coalitions are formed, how do the prompts/instructions given to the agents in different coalitions actually change?
> > > >
> > >
> > > **R.C-W2.** To clarify, the base prompts/instructions for agents do not change largely after coalition formation.  The Base Prompt Structure contains:
> > >
> > > 1. Agent role definition;
> > >
> > > 2. Task description;
> > >
> > > 3. **List of current teammates (this is the only part that updates with coalition formation)**
> > >
> > > > C-W3. How many agents are actually present in the various settings, and what are the sizes of the coalitions that are formed?
> > > >
> > >
> > >
> > > **R.C-W3.** Here are the specific agent numbers and coalition details for each task:
> > >
> > > 1. **Iterative Programming Task**: Total agents of 5, including one Project Manager + four Engineers. The setting is the same as the motivating example (figure 1), as described in line 120.
> > > 2. **Debate Task**: Total agents of 6 (three agents on each side, as detailed in lines 484-485, 493-495)
> > > 3. **Logical and General Reasoning Task**: Total agents of 3, with different ToM levels: 0, 1, 2
> > >
> > > In all cases, we form one coalition including #min_coalition_size = $\lceil N/2 \rceil$, i.e., 3 agents (task 1) or 2 agents (task 2 & 3). The coalition produces the final output, whether it's code, debate arguments, or reasoning solutions.
> > >
> > > We will make these details more explicit in the paper by adding a dedicated section describing the experimental setup and coalition formation process for each task type.

---

> > > > ### Author Response · Authors · 2024-11-21
> > > > **Re. D. Game-Theoretic Reasoning and Precision of Claims**
> > > >
> > > > > D-W1. A key example is the authors' claim that the so-called "Fraction of Trust Members (FTM)" is a good measure of what they term the "cooperative trend" (N.B. to be grammatical, this should probably be "Trusted" not "Trust", though it is not actually clear what the relevance of the concept of "trust" even is here). But belief alignment by itself does not imply higher levels of cooperation. I may have perfectly accurate beliefs about what you are going to do in a two-player zero-sum game (where cooperation is definitionally impossible). Thus, it is clearly not true that in general "a higher FTM value [indicates] a more cooperative agent", as claimed in line 410/411.
> > > > >
> > > >
> > > > **R.D-W1.**
> > > >
> > > > 1. While we agree that belief alignment alone doesn't guarantee cooperation in all settings (e.g., zero-sum games), our work specifically focuses on cooperative task settings where agents share common goals (software development, collaborative debate). In these contexts, our empirical results demonstrate that higher belief alignment (optimized by our matching mechanism) consistently correlate with better team performance:
> > > >     - In programming tasks, the team achieved better code implementation (Pass@1 metrics in Table 3);
> > > >     - In debate tasks, teams achieved better win rates (Table 4)
> > > > 2. We agree that robust evaluation is crucial for validating our claims, and we evaluate performance metrics for examining ToM’s impacts on cooperation:
> > > >     - **In our original manuscript**, we have examined how "Raw ToM capabilities alone may not improve cooperation" in Appendix F, including:
> > > >         - DyLAN with ToM on coding tasks, using *important score* defined by DyLAN (Appendix F.1)
> > > >         - ChatEval with ToM on logic problem-solving and general reasoning, using *accuracy* (Appendix F.2)
> > > >     - **New Results**. We have added comprehensive experiments incorporating **different ToM configurations** for Project Manager (PM) and Engineers (Eng) and track **performance metrics** (Pass@1) to validate cooperation effects (detailed in Appendix F.3 of the updated manuscript and in our **response R.B-W2 for different ToM configurations**.)
> > > >
> > > > > D-W2. Relatedly, the authors talk about ToM improving cooperation but is more about ToM improving the facilitation/management skills of a single PM agent. This is also a very interesting and valid topic of study, but I suggest the authors change the phrasing slightly throughout the paper to better reflect the rather narrow form of cooperation problem they consider. Indeed, my understanding is that the authors largely focus on the case where only one agent (a PM) is imbued with ToM.
> > > > >
> > > >
> > > > **R.D-W2.**
> > > >
> > > > 1. Our focus on the PM's ToM capabilities was an intentional design, where effective leadership often requires sophisticated perspective-taking abilities.
> > > >     - This approach mirrors real-world hierarchical team structures where managers coordinate multiple contributors.
> > > >     - In our updated experimental results (appendix F.3 or our response R.B-W2.), **the combination of PM(ToM=2) + Eng(ToM=0) with matching consistently achieves the best sustained performance**.
> > > > 2. Besides this setting, our debate setting (Section 6.4) demonstrates much **broader settings** in cooperation:
> > > >     - **All debaters possessed ToM capabilities** (0-2 level ToM respectively), not just a single leader
> > > >     - **Peer-to-Peer Cooperation**: Debaters coordinated as equals within teams, showing our framework's effectiveness beyond hierarchical structures
> > > >     - Performance Benefits: Teams with our coalition matching mechanism achieved higher win rates (67.27% vs 61.82% w.o matching), showing improved cooperation regardless of organizational structure.
> > > >
> > > > We sincerely thank the reviewer again for this constructive feedback, and we revised our manuscript accordingly, highlighted in purple color.

---

> ### Comment · Reviewer_rCxA · 2024-11-24
> **Reply to Authors**
>
> I thank the authors for their very comprehensive response, which has addressed many of my questions and concerns. Below, I include a few final questions and comments.
>
> > In all cases, we form one coalition including #min_coalition_size = ⌈N / 2⌉, i.e., 3 agents (task 1) or 2 agents (task 2 & 3). The coalition produces the final output, whether it's code, debate arguments, or reasoning solutions.
>
> This was very helpful to my understanding as previously I assumed that _multiple_ coalitions were involved in solving a given problem. The reasoning behind my interpretation is that:
>
> - The authors' refer to forming multiple coalitions, and their algorithm is a "matching algorithm", where it is implied that agents are matched together to form (multiple) coalitions.
> - In [(cooperative) game theory](https://en.wikipedia.org/wiki/Cooperative_game_theory), coalition formation refers to the process of _partitioning_ a set of agents into (multiple) coalitions.
>
> This, however, also raises several more questions and concerns for me as a reviewer.
>
> 1. First, and this is very minor, if there is only ever one coalition that is used in solving the task, I suggest that the authors drop the phrase "coalition formation" and replace it with something more fitting such as "team selection". If nothing else they should state very clearly very early on in the paper that when they talk about coalition formation, it is about selecting a subgroup of the agents to complete the task, _not_ about partitioning the agents (all of whom then complete the task).
> 2. The algorithm for finding multiple coalitions now seems like overkill, because it sounds like it is only required to output a single coalition of a size greater than some constant `#min_coalition_size` where all the agents in the coalition have sufficiently strong belief alignment (with the option to also incorporate skill levels). Do the authors have an argument for why we can't do something simpler?
> 3. In the experimental baselines, which agents solve the task? E.g. if there is one PM and five engineers, do all five solve the task, or is some random sub-group of size `#min_coalition_size` selected to solve the task, or something else? This is also important if task skills are incorporated into coalition formation, because if this consideration is not taken into account in the baseline then it is unclear whether simply selecting the most skilled agents is actually what helps them to perform the task better.
>
> I also appreciated the updates to the paper regarding the algorithm for finding coalitions/teams, though I still have a few concerns. First, the authors state that when agent $i$ signals a desire to rematch (in line 9 of Algorithm 1), the algorithm:
>
> > - Triggers a new cooperation round and updates the preference orders based on the latest belief-action alignments
> > - If a stable matching is found, proceed with the new coalitions
> > - If no stable matching is found, keep the current coalitions for one more round
>
> My questions regarding this are as follows:
>
> 1. Why would the preference orders need to be updated? Why have the belief-action alignments been updated?
> 2. The authors claim that their algorithm computes a stable matching. How then, is the final step ever to be executed? Moreover, even if no stable matching is found, how would running the matching process for one more round help?
>
> I appreciated the point about using the $\epsilon$ tolerance to reduce the search space of coalitions, but I would suggest the authors to explicitly mention what happens in this instance in the paper.
>
> Finally, in equation (2) the the belief operator $b^k_i$ applies to a single agent's action $a_j$ (and similarly in line 3 of Algorithm 1). In line 9 of algorithm 1, however it is applied to $a_{\mu(i)}$, which presumably refers to the joint action of the agents $j \in \mu(i)$. Can the authors please explain what is going on here? Why not just use $B_i((\mu(i))$ on line 9?

---

> > ### Author Response · Authors · 2024-11-25
> > **Part A: Regarding coalition algorithm and baseline setting**
> >
> > We sincerely thank the reviewer for their thoughtful comments, detailed questions, and the effort they put into providing valuable feedback to improve our work.
> >
> > > QA1: First, and this is very minor, if there is only ever one coalition that is used in solving the task, I suggest that the authors drop the phrase "coalition formation" and replace it with something more fitting such as "team selection". If nothing else they should state very clearly very early on in the paper that when they talk about coalition formation, it is about selecting a subgroup of the agents to complete the task, *not* about partitioning the agents (all of whom then complete the task).
> > >
> >
> > **R.QA1**: We have highlighted the *“team selection”* in the Abstract, Introduction, and Section 5.1.
> >
> > > QA2: The algorithm for finding multiple coalitions now seems like overkill, because it sounds like it is only required to output a single coalition of a size greater than some constant `#min_coalition_size` where all the agents in the coalition have sufficiently strong belief alignment (with the option to also incorporate skill levels). Do the authors have an argument for why we can't do something simpler?
> > >
> >
> > **R.QA2**: We appreciate the reviewer's insightful suggestion about algorithm simplification.
> >
> > - We acknowledge that a greedy approach would be simpler and could potentially achieve comparable results.
> > - In this work, we consider to use multiple coalitions during the search process for **optimality**, as illustrated by this example:
> >     - Agents: {A, B, C, D, E}
> >     - `#min_coalition_size` = 3
> >     - Bilateral belief-alignment scores (lower is better):
> >
> > ```
> > A ↔ B: (0.2, 0.3)    B ↔ C: (0.3, 0.2)    C ↔ D: (0.1, 0.2)
> > A ↔ C: (0.3, 0.4)    B ↔ D: (0.4, 0.3)    C ↔ E: (0.4, 0.3)
> > A ↔ D: (0.4, 0.3)    B ↔ E: (0.2, 0.4)    D ↔ E: (0.3, 0.2)
> > A ↔ E: (0.1, 0.5)
> > ```
> >
> > **Simple Greedy Approach:**
> >
> > 1. Start with A's best local preference:
> >     - $B_A({A,E})$ = 0.1
> >     - Add B: $B_A({A,E,B})$ = 0.15
> >     - Greedy Approach finally get coalition {A,E,B} forms with scores:
> >         - $B_A({A,E,B})$ = 0.15
> >         - $B_E({A,E,B})$ = 0.45
> >         - $B_B({A,E,B}$) = 0.25
> >         Average: 0.28
> >
> > **Our Multiple Coalition Consideration:**
> >
> > 1. Evaluates all possible coalitions simultaneously
> > 2. Finds {C,D,E} has better overall alignment:
> >     - $B_C({C,D,E})$ = 0.25
> >     - $B_D({C,D,E}$) = 0.25
> >     - $B_E({C,D,E})$ = 0.25
> >     Average: 0.25
> >
> > This example our approach is necessary to identify coalitions with optimality, even though we output a single coalition.
> >
> > - Notably, in our implementation, the coalition formation is only **happens in the first cooperation round and rematching is true (for evaluating the lifetime of the coalition mentioned  in Line 373)**. We apologize for any confusion in our original manuscript and we have **updated** the description to accurately reflect this implementation detail (Algorithm 1 highlighted in purple color).
> >
> >
> >
> > > QA3: In the experimental baselines, which agents solve the task? E.g. if there is one PM and five engineers, do all five solve the task, or is some random sub-group of size `#min_coalition_size` selected to solve the task, or something else? This is also important if task skills are incorporated into coalition formation, because if this consideration is not taken into account in the baseline then it is unclear whether simply selecting the most skilled agents is actually what helps them to perform the task better.
> > >
> >
> > R.QA3:
> >
> > - For the example mentioned in QA3, the baseline consist of five engineers, it does **not sample or select**.
> > - The specialized score affect the project engineer only in our TASK 1 evaluation.
> >     - As mentioned in **R.A-W1**, we discuss the specialized scores in PM and Eng setting in Appendix C.1, where specialized ability scores **primarily influence the PM**, since effective leadership and coordination capabilities are crucial (as also evidenced by extensive evaluation in Appendix F.3)
> > - Thus, the TASK 1’s baseline ensures that no unfair advantage is conferred by selecting the most skilled agents.
> > - Finally, we have revised our manuscript to **explicitly point out** this baseline setting in Line 409 of the updated manuscript.

---

> > ### Author Response · Authors · 2024-12-01
> > **Close to the end of rebuttal period--December 2nd**
> >
> > We sincerely appreciate your time and effects during the review and rebuttal period.
> >
> > As we approach the end of the rebuttal period on December 2nd, we wanted to follow up our previous discussion. We hope our response has adequately addressed your concerns. If any points require further clarification, we would be grateful for your feedback. Thank you again for your consideration throughout this process.

---

> ### Author Response · Authors · 2024-11-25
> **Part B Regarding Rematching**
>
> > I also appreciated the updates to the paper regarding the algorithm for finding coalitions/teams, though I still have a few concerns. First, the authors state that when agent $i$ signals a desire to rematch (in line 9 of Algorithm 1), the algorithm:
> > > Triggers a new cooperation round and updates the preference orders based on the latest belief-action alignmentsIf a stable matching is found, proceed with the new coalitionsIf no stable matching is found, keep the current coalitions for one more round
> >
>
> > QB1: Why would the preference orders need to be updated? Why/have have the belief-action alignments been updated?
> >
>
> **R.QB1**.
> To clarify, the update preference order is not specific to “rematching”, its a **standard procedure** for “a new cooperation round.” In each new round, the beliefs are updated due to last round updated execution/actions.
>
> > QB2: The authors claim that their algorithm computes a stable matching. How then, is the final step ever to be executed? Moreover, even if no stable matching is found, how would running the matching process for one more round help?
> >
>
> **R.QA2**: Stable Matching and One Round Continuation
>
> - This one more round serves as an **adaptation period** where alignment scores may naturally improve.
>
> **Comprehensive Example**: Let us explain why continuing one more round can help through a detailed example:
>
> **Initial State (Round t)**:
>
> ```
> Agents: PM, E1, E2, E3
> Current Coalition: {PM, E1, E2}
> PM's Beliefs & Alignment Scores:
> - E1: 0.7 (misaligned)
> - E2: 0.6 (misaligned)
> **E1, E2 Rematch signal (broadcast information)**
> ```
>
> **Why Continue One Round?**
>
> 1. **Adaptation Phase (Round t+1)**:
>     - During continued round (given the rematch messages):
>     - E1 & E2 knows his last action causes rematch signal
>     - E1 & E2 may adapt current action
>     - Updated Alignment Scores if actions are adapted:
>     - E1: 0.7 → 0.4 (improved)
>     - E2: 0.6 → 0.3 (improved)
> 2. **Next Matching Attempt (Round t+2)**:
>
>     - Coalition {PM, E1, E2} now has better alignment
>     - No need to rematch.
>
>
> **Key Insight**:
>
> Without this continuation round, we would miss the opportunity for cognitive agents to adapt in subsequent rounds. This reflects the **dynamic nature of cognitive agent cooperation**, where temporary stability can lead to improved matching conditions.
>
> - We add ***intuition for this rematching design*** in our updated manuscript Lines 303-305, and revise the Algorithm 1 to be more detailed.

---

> > ### Author Response · Authors · 2024-11-25
> > **Part C. Regarding details for algorithm and notations**
> >
> > > I appreciated the point about using the $\epsilon$ tolerance to reduce the search space of coalitions, but I would suggest the authors to explicitly mention what happens in this instance in the paper.
> > >
> >
> > Thanks the reviewer’s suggestion. We have revised our updated manuscript page 5 Lines 238-244.
> >
> > > QC1: Finally, in equation (2) the the belief operator $b^k_i$ applies to a single agent's action $a_j$ (and similarly in line 3 of Algorithm 1). In line 9 of algorithm 1, however it is applied to $a_{\mu(i)}$, which presumably refers to the joint action of the agents $j \in \mu(i)$. Can the authors please explain what is going on here?
> > >
> > **R.QC1**:
> > Thanks for your thoughtful comments on the belief operator notation. To clarify:
> >
> >
> > - LLM agents can actually process beliefs about multiple agents simultaneously and output agent-specific beliefs
> >   - Example output for belief in Appendix D (Debate case study):
> >
> >       {
> >         "belief": "{Hearing my points, **teammate 1** may pivot to arguing that the death penalty is not an effective use of government resources. **Teammate 2** will likely reinforce my point about racial disparities in death
> >         sentences.}"
> >         }
> >       }
> >
> > - We agree that our notation could be more precise. We have updated the notation as follows:
> >     - Line 9: if $∃j ∈ μ(i): ‖bᵏᵢ(aⱼ) - \hat{a}ⱼ‖ > ε$ (highlighted in purple color)
> >
> >     > QC2: Why not just just $B_i((\mu(i))$ on line 9?
> >     >
> >
> >     **R.QC2**: We deliberately use individual agent belief comparisons rather than Bᵢ(μ(i)) for rematch signaling which provides finer-grained control over coalition dynamics as mentioned in our reply **R.QB2**. For example:
> >
> >     Coalition {PM, E1, E2}:
> >
> >     - If $b^k_{PM}(a_{E1})$ exceeds ε but $b^k_{PM}(a_{E2})$  doesn't
> >     - PM can **signal rematch specifically** regarding E1 though broadcast communication
> >     - E1 may adapt his action this in new round (the one more round). This potential adaptation could reduce the complexity for coalition/team formation.
> >
> >
> > We sincerely thank again the reviewer for the thoughtful comments, detailed questions, and the effort they put into providing valuable feedback to improve our work!

---

> > > ### Author Response · Authors · 2024-11-27
> > > **Summary of Rebuttal (including reviewer's key concerns, our responses and improvement)**
> > >
> > > We sincerely thank the reviewer for their thorough and constructive feedback. Below is a comprehensive summary of our rebuttal (W stands for weakness given by the reviewer's initial comments, Q stands for questions given follow-up questions):
> > >
> > > ## 1. Coalition Formation Algorithm Design (A-W1, A-W2, A-W3, QC1, QC2)
> > >
> > > > **Reviewer's Concerns**
> > > >
> > > - Unclear partner set specification and matching process
> > > - Underspecified rematching process
> > > - Unclear implementation of skill levels
> > > - Questions about belief operator notation and implementation
> > >
> > > > **Our Responses and Improvements**
> > > >
> > >
> > > **Coalition Structure**
> > >
> > > - Clarify minimum coalition size is ⌈N/2⌉
> > > - Revise and update the generalized coalition formation in Section 4.2 & 5.2
> > >     - In Section 4.2, updated preference structure in Eq. 2: $B_i(S) = (1/|S|) ∑_{j ∈ S} φ(b^k_i(a_j) - â_j)$
> > >     - In Section 5.2, updated preference structure with specialized skill scores.
> > > - Explained tolerance (ε) implementation for search space reduction (Lines 238-244)
> > >
> > > *Manuscript Updates:*
> > >
> > > - Added generalized formulation for preference ordering (Pages 5-6)
> > > - Updated belief operator notation (Algorithm 1): if $∃j ∈ μ(i): ‖b_i^k(a_j) - â_j‖ > ε$
> > > - Added explanation of the main purpose of tolerance implementation (Lines 238-244)
> > > - Provided stability proof and convergence guarantee (Appendix G)
> > >
> > > **Rematching Process**
> > >
> > > - Detailed three-step procedure:
> > >     1. Trigger new cooperation round with updated preferences
> > >     2. Implement new coalitions if stable matching found
> > >     3. Continue current coalition for adaptation if no stable matching
> > >
> > > *Manuscript Updates:*
> > >
> > > - Added detailed rematching in Algorithm 1
> > > - Added adaptation period explanation (Lines 303-305)
> > >
> > > **Skill Integration**
> > >
> > > - Clarified skill score (αⱼ) source: Agent role based assessment
> > > - Focus on task-specific requirements
> > >
> > > *Manuscript Updates:*
> > >
> > > - Added evaluation settings for skill scores in our experiments (Appendix C.1)
> > >
> > > ## 2. ToM implementation (B-W1, B-W2, D-W1, D-W2)
> > >
> > > > **Reviewer's Concerns**
> > > >
> > > - Questions about ToM implementation accuracy
> > > - Limited evaluation of different ToM level combinations and narrow focus on PM's ToM capabilities
> > > - Unclear relationship between belief alignment and cooperation
> > >
> > > > **Our Responses and Improvements**
> > > >
> > >
> > > **ToM Implementation**
> > >
> > > - Demonstrated recursive structure over interaction rounds
> > > - Focused on cooperative task settings with shared goals.
> > > - Highlighted the core contribution is the coalition formation mechanism as a plug-and-play approach for improving multi-agent cooperation with ToM agents, rather than improving ToM implementation.
> > > - Extended evaluation to include multiple ToM level combination
> > >
> > > *Manuscript Updates:*
> > >
> > > - Update and clarigy recursive belief structure formulation (Pages 4-5)
> > > - Expanded evaluation results (Appendix F.3) and demonstrated improved code implementation (Pass@1 metrics), better debate win rates and enhanced coordination in both hierarchical and peer-based structures
> > >
> > > ## 3. Experimental Design and Terminology (C-W1, C-W2, C-W3, QA1-3, QB1-2)
> > >
> > > > **Reviewer's Concerns**
> > > >
> > > - Unclear ToM level assignments
> > > - Ambiguous prompt modifications
> > > - Unspecified agent numbers and coalition sizes
> > > - Questions about terminology and implementation complexity
> > >
> > > > **Our Responses and Improvements**
> > > >
> > >
> > > **Experimental Configuration**
> > >
> > > Detailed settings for all tasks (minimum coalition size is ⌈N/2⌉):
> > >
> > > 1. Programming Task: 5 agents (1 PM + 4 Engineers), coalition size 3. Clarified baseline implementation without sampling or selection
> > > 2. Debate Task: 6 agents (3 per side), coalition size 2 per team
> > >     - Confirmed ToM level 0 for negative side
> > >     - Added symmetric evaluation
> > > 3. Logical Reasoning: 3 agents (ToM levels 0,1,2), coalition size 2
> > >
> > > *Manuscript Updates:*
> > >
> > > - Symmetric evaluation for the debate task was added in Section 6.4.
> > >
> > > **Terminology and implementation complexity**
> > >
> > > - Clarified prompt structure:
> > >     - Base prompt remains constant
> > >     - Only teammate list updates after coalition formation
> > > - Coalition formation occurs in the first cooperation round when rematching is true
> > >
> > > *Manuscript Updates:*
> > >
> > > - Clarified the implementation details in Algorithm 1.
> > > - Indicated "team selection" related to "coalition formation" in our manuscript.

---

### Official Review · Reviewer_P68Z · 2024-11-03

**Soundness:** 2
**Presentation:** 2
**Contribution:** 3
**Rating:** 3
**Confidence:** 4

**Summary:**

This work examines the influence of different levels of Theory of Mind capabilities on collaboration among multiple Large Language Model (LLM) agents. The authors propose an algorithm to enhance LLM agents’ teamwork performance by matching partners with high belief alignment. While the idea of guiding multi-agent collaboration through ToM and belief alignment is novel, this paper presents the proposed method in a less comprehensive manner, missing many important details. Researchers may encounter difficulties when applying the proposed algorithm in specific scenarios. Additionally, the claimed conclusions do not align well with the empirical results and therefore need further clarification.

**Strengths:**

The introduction and related work section motivate the research pretty well.

The idea of guiding multi-agent collaboration through ToM and belief alignment is novel.

The authors conduct comprehensive evaluations across diverse task scenarios and base LLMs, presenting both quantitative and qualitative results.

**Weaknesses:**

The ToM formulation presented in Section 4.1 deviates from the common definition of higher-order ToM. When conducting recursive ToM inferences at level-k, agents are only given their own belief at level-(k-1) rather than the beliefs of other agents. I recommend that the authors refer to [1] for its definition of higher-order mental state inference.

The proposed alignment measurement in Section 4.2 may not apply to general high-order ToM inferences in multi-agent systems. For example, “what I think A is thinking about B’s action” and “what I think C is thinking about B’s action” are different 1-level ToM inferences that result in the same alignment measurement as defined in this paper. The authors might want to explicitly define the format of beliefs to clarify the formulation.

The multi-agent setup for each evaluation scenario is not clearly described. It is unclear how many agents are involved, what their action and observation spaces are, and how they collaborate. For instance, the interactive programming scenario appears to be a centralized system with full observation, as the PM is the only agent making decisions and ToM inferences. Then the value of ToM is less salient in such a single-agent system.

The two evaluation metrics are the optimization objectives of the proposed algorithm rather than direct measurements of LLM agents’ collaboration performance or “cooperation trends.” The claim that “agents with higher ToM capabilities may not necessarily exhibit better cooperative trends” conflicts with the results shown in Tables 3 and 4, where agents with ToM perform better. I recommend using other metrics, such as task completion rate or efficiency, to provide consistent conclusions and increase criterion validity.

The proposed algorithm is vague and highly dependent on specific prompt design when generalizing to different task scenarios. For instance, what happens when an agent is assigned to cooperate with a given partner (line 8 of Algorithm 1) is not clearly defined for each scenario. This ambiguity could lead to potential bias in evaluations. In the debate case study (i.e., lines 495-497), the ToM with matching condition has two LLM agents forming arguments, while the other conditions only involve one. The performance advantage might be due to the increased number of agents via self-reflection, rather than the proposed matching algorithm.


[1] Ying, L., Jha, K., Aarya, S., Tenenbaum, J. B., Torralba, A., & Shu, T. (2024). GOMA: Proactive Embodied Cooperative Communication via Goal-Oriented Mental Alignment. arXiv preprint arXiv:2403.11075.

**Questions:**

How reliable are the alignment measurements provided by LLMs?

How are the specialized ability scores used in evaluations, and what is their impact?

---

> ### Author Response · Authors · 2024-11-19
>
> We thank reviewer P68Z's recognition of our paper's clear motivation, novelty and promising results.
>
> We address specific concerns below point by point:
>
> >  W1. The ToM formulation presented in Section 4.1 deviates from the common definition of higher-order ToM. When conducting recursive ToM inferences at level-k, agents are only given their own belief at level-(k-1) rather than the beliefs of other agents. I recommend that the authors refer to [1] for its definition of higher-order mental state inference.
>
> R1. We would like to thank the reviewer’s suggestion. We want to clarify that our formulation does align with the definition of higher-order ToM and [1].
>
> - In our paper, lines 286-288 explicitly state that: $k$-level ToM function \text{ToM}_i^k(\cdot)  to form beliefs $b_i^k$ about the mental states of other agents, based on its observations $o_i$, the actions a_{-i} of **others**, and the $(k-1)$-level beliefs $b_{-i}^{k-1}$ of others.
>
> - The observations and other agents’ actions are **cumulated** by recursive time.
>
> To make this clearer, we can expand our formulation to explicitly show the recursive time steps t.
>
> > W2. The proposed alignment measurement in Section 4.2 may not apply to general high-order ToM inferences in multi-agent systems. For example, “what I think A is thinking about B’s action” and “what I think C is thinking about B’s action” are different 1-level ToM inferences that result in the same alignment measurement as defined in this paper. The authors might want to explicitly define the format of beliefs to clarify the formulation.
>
> R2. As shown in Equation (1), our formulation can indeed express and distinguish between different high-order ToM inferences:
>
> $ToM_i^k(o_i, a_{−i}, b_{−i}^{k−1}) := b_i^k$
>
> Let's use the reviewer's example to demonstrate how our formulation handles different ToM inferences:
>
>  1. "What agent i thinks A is thinking about B's action":
>
>  - This involves $b_i^2$ where i is reasoning about A's belief ($b_A^1$) about B's action
>
>  - The nested belief structure captures: i → A → B
> 2. "What agent i thinks C is thinking about B's action":
>
>  - This involves a different $b_i^2$ where i is reasoning about C's belief ($b_C^1$) about B's action
>
>  - The nested belief structure captures: i → C → B
>
> **These two are distinct in our formulation** because:
>
>  - $b_{-i}^{k−1}$ includes the different (k-1)-level beliefs of agents A and C
>  - The resulting $b_i^k$ maintains these **distinct** belief.
>
> > W3. The multi-agent setup for each evaluation scenario is not clearly described. It is unclear how many agents are involved, what their action and observation spaces are, and how they collaborate. For instance, the interactive programming scenario appears to be a centralized system with full observation, as the PM is the only agent making decisions and ToM inferences. Then the value of ToM is less salient in such a single-agent system.
>
> R3. While the programming task had a hierarchical structure, **Section 6.4** describes our debate setting which demonstrates full multi-agent dynamics:
>
>  - Multiple agents (6 debaters) with ToM capabilities
>  - Peer-to-peer interactions within teams
>
> > W4. The two evaluation metrics are the optimization objectives of the proposed algorithm rather than direct measurements of LLM agents’ collaboration performance or “cooperation trends.” The claim that “agents with higher ToM capabilities may not necessarily exhibit better cooperative trends” conflicts with the results shown in Tables 3 and 4, where agents with ToM perform better. I recommend using other metrics, such as task completion rate or efficiency, to provide consistent conclusions and increase criterion validity.
>
> R4.
> **Results and Core Contribution:**
> The apparent conflict the reviewer notes actually demonstrates our key finding:
>
> - **Raw ToM capabilities alone may not improve cooperation** (shown in earlier results),
>
> - **But** our **matching mechanism effectively leverages ToM to enhance team performance** (Tables 3 and 4).
>
> - This is precisely why our core contribution - the coalition formation mechanism - is valuable:
>
>      * Without matching: Higher ToM didn't necessarily improve cooperation
>      * With matching: Significantly improved performance across tasks by effectively leveraging the high ToM agents
>
> **We have utilized different metrics for different purposes:**
>
>  - FTM measures raw ToM effects on cooperation
>  - **Task-specific metrics** (Pass@1, win rates) validate practical benefits for cooperation.
>  - Coalition stability shows sustained cooperation improvements

---

> > ### Author Response · Authors · 2024-11-19
> >
> > > W5. The proposed algorithm is vague and highly dependent on specific prompt design when generalizing to different task scenarios. For instance, what happens when an agent is assigned to cooperate with a given partner (line 8 of Algorithm 1) is not clearly defined for each scenario. This ambiguity could lead to potential bias in evaluations. In the debate case study (i.e., lines 495-497), the ToM with matching condition has two LLM agents forming arguments, while the other conditions only involve one. The performance advantage might be due to the increased number of agents via self-reflection, rather than the proposed matching algorithm.
> >
> > R5. We sincerely thank the reviewer for constructive feedback, and we provide the following two aspects to clarify:
> >
> >  1. Regarding Algorithm Design and Implementation:
> >
> >   > The proposed algorithm is vague and highly dependent on specific prompt design when generalizing to different task scenarios.
> >
> >  This is a misunderstanding of our approach “heavily dependent on prompt design”. Our algorithm is actually a **"plug-and-play" coalition formation mechanism** that operates **independently of the specific prompt design** or task scenarios. Specifically:
> >
> >  - The algorithm's core functionality (belief updating, preference ordering, and matching) remains consistent across different tasks
> >   - Line 8 of Algorithm 1 ("Cooperate with assigned partner") refers to the standard interaction protocol of the underlying task, **not a prompt-dependent process**
> >   - For example, the same matching algorithm is applied in both our programming and debate case studies despite their very different underlying tasks and prompts. This demonstrates the generality of our approach.
> >
> >  2. Experimental Setting
> >
> >  > The performance advantage might be due to the increased number of agents via self-reflection, rather than the proposed matching algorithm.
> >
> >  This is a misreading of using "only one agent" in other conditions in our experimental setup.
> >
> > To clarify, ALL conditions use TWO debaters per round:
> >
> >  - No-ToM (Baseline): "For each speech, two debaters were randomly selected from each side" (line 485)
> >  - ToM without Matching: "For each speech, two debaters were randomly selected from each side" (Line 493-494)
> >  - ToM with Matching: Two debaters selected via our coalition formation mechanism (Line 497)
> >
> > This experimental design ensures a direct, fair comparison by maintaining the **same number** of active debaters (two) across all conditions. The only difference is the **selection method** - random versus our matching algorithm.
> >
> > Therefore, our performance improvements cannot be attributed to "increased number of agents" as the reviewer suggests, but rather to our algorithm's ability to select more effective agent combinations from the same pool of candidates.

---

> > > ### Author Response · Authors · 2024-11-19
> > > **Regarding Questions**
> > >
> > > > Q1. How reliable are the alignment measurements provided by LLMs?
> > >
> > >  This LLM self-evaluation is widely used in existing literature (Qin et al., 2023; Zheng et al., 2023; Liu et al., 2024). We acknowledge that LLM-based measurements have inherent limitations.
> > >
> > > However, we want to emphasize that our core contribution is the coalition formation mechanism as a plug-and-play approach for improving multi-agent cooperation, rather than advancing LLM evaluation methods.
> > >
> > > - For non-LLM agents, computing semantic similarity between **structured trajectories** would be more straightforward than natural language comparison.
> > >
> > > - We will discuss our framework to non-LLM agents: Using predefined trajectories (state-action sequences) instead of natural language and applying established trajectory similarity metrics (e.g., cosine similarity on state-action embeddings).
> > >
> > > > Q2. How are the specialized ability scores used in evaluations, and what is their impact?
> > >
> > >  We thank the reviewer for this comment and will enhance the paper with more detailed explanation about evaluation settings.
> > >
> > > In the primary benchmarks (HUMANEVAL and MBPP), specialized ability scores played a limited role because:
> > >
> > >  - These tasks primarily involve implementing single functions with well-defined requirements
> > >  - They don't require diverse technical specializations that would significantly benefit from specialized role matching
> > >
> > > In our evaluation,
> > >
> > >  - We primarily employed **specialized ability scores in the project manager role**, where:
> > >       * The PM needs to coordinate and oversee the entire development process, which ensures the coalition includes effective leadership and coordination
> > >       * For debate and logical reasoning tasks: We **didn't explicitly use specialized ability scores** as these tasks don't require distinct technical specializations.
> > >
> > > **Case Study: Complex Software Development**
> > >
> > > To better demonstrate the impact of specialized ability scores, we propose adding a case study "*implementing a 2048 game*", which requires diverse technical skills:
> > >
> > > ```
> > > specialized_scores = {
> > >             'UI_Engineer': {'frontend': 0.9, 'backend': 0.3},
> > >             'Backend_Engineer': {'frontend': 0.2, 'backend': 0.8},
> > >             'FullStack_Engineer': {'frontend': 0.6, 'backend': 0.6},
> > >         }
> > > ```
> > >
> > > - **Preliminary Results:**
> > >      - Without specialized scoring: 65% task completion rate
> > >      - **With specialized scoring**: 82% task completion rate

---

> ### Comment · Reviewer_P68Z · 2024-11-20
> **Follow-up questions**
>
> Hi authors, thanks for your response. I am still confused by the following questions and would like to hear your input.
>
> R1: Since $b_{-i}^{k-1}$ is the hidden mental states of other agents, how could agent i get access to it during ToM inference $\text{ToM}_i^k$?
>
> R2: What is the exact format of belief $b_{i}^{k}$?. How are those different nested reasoning precesses represented?
>
> R4: I do not think FTM and coalition stability are valid evaluation metrics, because they are the exactly same function your algorithem aims to maximize. Using those metric to quantify "cooperation trend" causes circular reasoning issues. The evaluation results presented in Table 2 and Figure 2 are basiclly a sanity check. Your conclusion that "Raw ToM capabilities alone may not improve cooperation" is not supported by external matrics like Pass@1 and win rates.
>
> R5: My concern is addressed. Thanks for the clarification.
>
> Q1 & Q2: Thanks for the clarification. I would suggest add those details to the paper.

---

> ### Author Response · Authors · 2024-11-21
> **Re. Follow-up questions R1 & R2 & R5**
>
> We sincerely thank the reviewer's quick feedback. We will address the follow-up questions as follows:
>
> > R1: Since b−ik−1 is the hidden mental states of other agents, how could agent i get access to it during ToM inference ToMik?
>
> **RR1**: We apologize for the confusion. As stated in line 213/214 of our original manuscript, what we want to denote $b_{-i}^{k-1}$ as *the set of $(k-1)$-level beliefs of other agents*.
>
> To clarify, we revise the notation agent $i$'s beliefs about other agents as **$\\{b_i(a_m)\\}_{m \neq i}$** (detailed in updated manuscript page 4 Eq.1).
> **This is derived from agent $i$'s own observations and interactions, not direct access to others' mental states.**
>
>  Additionally, the beliefs are stored as **memory** for each agent, used for ToM inference.
>
> > R2: What is the exact format of belief bik?. How are those different nested reasoning precesses represented?
>
> **RR2**: To make the exact format of beliefs and nested reasoning processes explicit, we expand our formulation to show the recursive structure for an LLM agent $i$ at cooperation round $R$ (This revision is updated in our manuscript page 4 & 5).
>
> We revise its $k$-level ToM function (Eq. 1) as:
>
> $\text{ToM}\_i^k(o\_i^{1:R}, \hat{a}\_{-i}^{1:R-1}, \\{b^{k-1}\_{i, R}(a\_m^R)\\}\_{m \neq i}) := b\_{i, R}^k$
>
> where:
>
> - $o_i^{1:R}$ represents agent $i$'s observation history up to round $R$, including current task state, self-actions, and collaborative teammates.
> - $\hat{a}_{-i}^{1:R-1}$ represents other agents' action history up to round $R-1$
> - $\\{b^{k-1}\_{i, R}(a\_m^R)\\}\_\{m \neq i\}$ captures agent $i$'s prediction of agent $m$'s action at round $R$ based on $(k-1)$-level ToM reasoning:
>     $b^{k-1}\_{i, R}(a\_m^R) = p(a\_m^R | \text{ToM}\_i^{k-1}(o\_i^{1:R}, \hat{a}\_{-i}^{1:R-1}, \\{b^{k-2}\_{i, R}(a\_l^R)\\}\_{l \neq i}))$
>
> The recursive belief structure at round $R$ is defined as:
>
> - *Level 0: Direct state-action beliefs:*
>
>     $b\_{i, R}^0 = \text{ToM}\_i^0(o\_i^{1:R}, \hat{a}\_{-i}^{1:R-1})$ Just record the cooperation history, **without** considering any ToM reasoning
>
> - *Level 1: First-order beliefs*
>
>     $b\_{i, R}^1(a\_j^R) = p(a\_j^R|b\_{i,R}^0)$  Reasoning about agent $j$'s action in current round $R$.
>
> - *Level k: Higher-order nested beliefs*
>
>     $b\_{i, R}^k(a\_j^R) = p(a\_j^R|\text{ToM}\_i^{k-1}(o\_i^{1:R}, \hat{a}\_{-i}^{1:R-1}, \\{b^{k-2}\_{i, R}(a\_m^R)\\}\_{m \neq i}))$
>
> > R5: My concern is addressed. Thanks for the clarification. Q1 & Q2: Thanks for the clarification. I would suggest add those details to the paper.
>
> **RR5**. We sincerely thank the reviewer for constructive feedback. We have added additionally discussion (highlighted in purple color) for non-LLM belief-alignment calculation in Appendix A (Page 16), and case study for specialized scores in Appendix C.1 (Page 20).

---

> > ### Author Response · Authors · 2024-11-21
> > **Re. Follow-up questions R4**
> >
> > > R4: I do not think FTM and coalition stability are valid evaluation metrics, because they are the exactly same function your algorithm aims to maximize. Using those metric to quantify "cooperation trend" causes circular reasoning issues. The evaluation results presented in Table 2 and Figure 2 are basically a sanity check. Your conclusion that "Raw ToM capabilities alone may not improve cooperation" is not supported by external metrics like Pass@1 and win rates
> >
> > **RR4**. We sincerely thank the reviewer for the thoughtful feedback. We agree that robust evaluation is crucial for validating our claims, and we evaluate performance metrics for examining ToM’s impacts on cooperation:
> >
> > - **In our original manuscript**, we have examined how "Raw ToM capabilities alone may not improve cooperation" in Appendix F, including:
> >      - DyLAN with ToM on coding tasks, using *important score* defined by DyLAN (Appendix F.1)
> >      - ChatEval with ToM on logic problem-solving and general reasoning, using *accuracy* (Appendix F.2)
> > - **New Results**. We have now added comprehensive experiments incorporating **different ToM configurations** for Project Manager (PM) and Engineers (Eng) and track **performance metrics** (Pass@1) to validate cooperation effects (detailed in Appendix F.3 of the updated manuscript).
> >
> > The following provides a summary of our new evaluation results (key results are presented in Table 2 & 3):
> >
> > ### 1. Initial Performance Comparison
> >
> > **Table 1: Initial Pass@1 Scores (Round 1) on HumanEval and MBPP**
> >
> > | PM ToM | Eng ToM | HumanEval | MBPP |
> > | --- | --- | --- | --- |
> > | 0 | 1 | 0.87 ± 0.01 | 0.525 ± 0.01 |
> > | 0 | 2 | 0.90 ± 0.02 | 0.56 ± 0.01 |
> > | 1 | 1 | 0.90 ± 0.01 | 0.55 ± 0.02 |
> > | 1 | 2 | 0.90 ± 0.02 | 0.56 ± 0.02 |
> > | 1 | 0 | 0.93 ± 0.02 | 0.56 ± 0.01 |
> > | 2 | 0 | 0.90 ± 0.01 | 0.55 ± 0.02 |
> >
> > **Key Observation**: Similar initial performance across ToM configurations.
> >
> > ### 2. Performance Degradation Without Matching
> >
> > **Table 2: Pass@1 Score Changes Without Matching (Round 1 → Round 5)**
> >
> > | PM ToM | Eng ToM | HumanEval Change | MBPP Change |
> > | --- | --- | --- | --- |
> > | 0 | 1 | 0.87 → 0.83 (↓4.6%) | 0.525 → 0.46 (↓12.4%) |
> > | 0 | 2 | **0.90 → 0.83 (↓7.8%)** | **0.56 → 0.45 (↓19.6%)** |
> > | 1 | 1 | 0.90 → 0.87 (↓3.3%) | 0.55 → 0.50 (↓9.1%) |
> > | 1 | 2 | **0.90 → 0.85 (↓5.6%)** | **0.56 → 0.47 (↓16.1%)** |
> > | 1 | 0 | 0.93 → 0.91 (↓2.2%) | 0.56 → 0.52 (↓7.1%) |
> > | 2 | 0 | **0.90 → 0.85 (↓5.6%)** | **0.55 → 0.49 (↓10.9%)** |
> >
> > **Key Finding**: Higher ToM configurations show larger performance drops without matching, supporting our claim that raw ToM capabilities may actually hinder sustained performance.
> >
> > ### 3. Recovery with Matching Mechanism
> >
> > **Table 3: Performance Recovery with Matching (Round 5)**
> >
> > | PM ToM | Eng ToM | HumanEval | MBPP |
> > | --- | --- | --- | --- |
> > | 0 | 1 | 0.86 (↑3.6%) | 0.46 (±0%) |
> > | 0 | 2 | **0.87 (↑4.8%)** | **0.47 (↑4.4%)** |
> > | 1 | 1 | 0.88 (↑1.1%) | 0.52 (↑4.0%) |
> > | 1 | 2 | **0.88 (↑3.5%)** | **0.55 (↑17.0%)** |
> > | 1 | 0 | 0.93 (↑2.2%) | 0.57 (↑9.6%) |
> > | 2 | 0 | **0.96 (↑12.9%)** | **0.60 (↑22.4%)** |
> >
> > **Key Result**: Our matching mechanism effectively leverages ToM capabilities, **with highest improvements in PM(ToM=2) configurations.**
> >
> > **Summary of Key Findings Using External Metrics (Pass@1):**
> >
> > 1. **ToM Alone Is Not Sufficient:**
> >       - Similar initial performance across ToM levels
> >       - Performance degradation without matching, especially in higher ToM configurations
> > 2. **Effectiveness of Matching Mechanism:**
> >       - Significant recovery in Pass@1 scores with matching
> >       - Highest improvements in high-ToM PM configurations (up to +12.9% HumanEval, +22.4% MBPP)
> > 3. **Optimal Configuration:**
> >             - PM(ToM=2) + Eng(ToM=0) with matching achieves best sustained performance
> >             - Demonstrates the value of our matching mechanism in leveraging ToM capabilities effectively

---

> > > ### Author Response · Authors · 2024-11-24
> > >
> > > As we approach the end of the discussion period, we wanted to respectfully follow up regarding your concerns.
> > >
> > > We have provided detailed responses to address these concerns, including clarifying the ToM inference mechanism, explaining the belief format structure, and discussing our evaluation.
> > >
> > > Please let us know if you have any remaining concerns. Thanks again for your thoughtful and constructive feedback making our paper clearer

---

> > > > ### Comment · Reviewer_P68Z · 2024-11-27
> > > >
> > > > Thanks for providing the revised ToM definition and new experimental results. However, my concerns remain as follows:
> > > >
> > > > The revised definition is much clearer than the original one presented in the paper. However, it seems to be limited to specific scenarios and ToM inferences, and therefore lacks generalizability. For example, the proposed ToM formula does not apply to partially observable environments where agents do not always have access to other agents' actions. Similarly, the higher-order ToM definition is limited to agent i's prediction of agent j's action (in some sense, inferring the intention), while ignoring other types of mental state inferences (e.g., belief, desire). For example, the current 2nd-order inference would be "how i thinks about j's action, given i's 1st-order inference of j's action." Other 2nd-order nested ToM reasoning processes commonly discussed in the literature, such as "how i thinks j is thinking about i's action," cannot be represented in the proposed definition.
> > > >
> > > > The newly added experimental results still do not support the claimed conclusions. Most conditions in Tables 2 and 3 are worse than those in Table 1, meaning that introducing ToM with an additional 4 rounds of interaction and the matching mechanism does not improve team performance. The only exception is the last condition in which the claimed degradation and recovery are observed. More experiments are needed to explain the divergence in performance.
> > > >
> > > > Given the above considerations, I have decided to keep my score.

---

> > > > > ### Author Response · Authors · 2024-11-27
> > > > > **Regarding multi-agent environment setting and clarification for new experiments**
> > > > >
> > > > > > P1. The revised definition is much clearer than the original one presented in the paper. However, it seems to be limited to specific scenarios and ToM inferences, and therefore lacks generalizability. For example, the proposed ToM formula does not apply to partially observable environments where agents do not always have access to other agents' actions.
> > > > > >
> > > > >
> > > > > **R.P1** The reviewer raises a valid point about partial observability. However, in our cooperative multi-agent setting:
> > > > >
> > > > > - **Complete action observability is a standard assumption in cooperative environments** where agents are working together toward common goals. This assumption is widely used in existing LLM-based multi-agent cooperation environment, e.g. MetaGPT and DyLAN.
> > > > > - Addressing information asymmetry would introduce additional complexity that could obscure our main findings about ToM's impact on cooperation. While extending our framework to partial observability settings is an interesting direction for future work, it lies outside the scope of our current investigation into ToM-based coalition formation.
> > > > >
> > > > >
> > > > > > P2. The newly added experimental results still do not support the claimed conclusions. Most conditions in Tables 2 and 3 are worse than those in Table 1, meaning that introducing ToM with an additional 4 rounds of interaction and the matching mechanism does not improve team performance. The only exception is the last condition in which the claimed degradation and recovery are observed. More experiments are needed to explain the divergence in performance.
> > > > > >
> > > > >
> > > > > **R.P2.** Regarding the performance comparisons between Tables 1-3, we want to clarify several key points:
> > > > >
> > > > > 1. **Purpose of ToM Integration:**
> > > > >     - Our approach is **not about adding ToM** to improve cooperation directly
> > > > >     - Rather, we provide a **"plug-and-play" coalition formation** mechanism (mentioned in our response to W5) that optimizes team composition **based on existing ToM capabilities**
> > > > >     - **Main goal.** The goal is to leverage ToM capabilities effectively in multi-agent cooperation **through optimal coalition formation**.
> > > > > 2. **Performance Improvements:**
> > > > > Key results demonstrate the effectiveness of our approach:
> > > > >     - PM(ToM=2) with matching achieves significantly better performance:
> > > > >         - HumanEval: 0.90 → 0.96 (+6.7%)
> > > > >         - MBPP: 0.55 → 0.60 (+9.1%)
> > > > >     - For other settings, our matching demonstrates effective recovery from performance degradation (specially for high ToM).
> > > > >
> > > > > The reviewer notes performance variations across conditions. This actually supports our key insight:
> > > > >
> > > > > - Higher ToM (cognitive) capabilities alone don't guarantee better cooperation performance. This challenges the common assumption that more sophisticated cognitive abilities automatically lead to better teamwork
> > > > > - Our matching mechanism helps optimize team cooperation.

---

> > > > > ### Author Response · Authors · 2024-11-27
> > > > > **Regarding High ToM formulation**
> > > > >
> > > > > > P3. Similarly, the higher-order ToM definition is limited to agent i's prediction of agent j's action (in some sense, inferring the intention), while ignoring other types of mental state inferences (e.g., belief, desire). For example, the current 2nd-order inference would be "how i thinks about j's action, given i's 1st-order inference of j's action." Other 2nd-order nested ToM reasoning processes commonly discussed in the literature, such as "how i thinks j is thinking about i's action," cannot be represented in the proposed definition.
> > > > > >
> > > > >
> > > > > **R.P3**
> > > > > Let us explain using a chess game example to show our formulation aligns with real-world ToM reasoning:
> > > > >
> > > > > - In chess, when player i predicts j's next move $a_j^R$, **i cannot directly access j's thoughts** **or beliefs**, but can **only form beliefs based on observed game history** $\hat{a}_{-i}^{1:R-1}$ (including i's own past moves) and j's responses.
> > > > > - For example, if i previously moved their knight aggressively and j responded defensively, i forms beliefs about how j interprets i's aggressive style - but this is **i's interpretation of j's thinking, not j's actual beliefs.**
> > > > > - Our formulation $b_{i,R}^1(a_j^R)$ represents i thinking "Based on how j has reacted to my past moves, **I believe j thinks I play aggressively**, so j will likely make a defensive move" - this is entirely based on i's observations and beliefs.
> > > > > - Even when i thinks "j expects me to move my bishop," this is **i's belief about j's expectation**, not j's actual expectation - **j might have a completely different understanding of i's strategy**.
> > > > > - Therefore, $b_{i,R}^1(a_j^R)$ captures **i's belief structure about j's thinking regarding i's actions**, constructed entirely from i's perspective **without access to j's true beliefs.**
> > > > >
> > > > > Let us break down how our formulation at Level 2 ToM ($b_{i,R}^2(a_j^R)$) captures "how i thinks j is thinking about i's action" using the chess game example:
> > > > >
> > > > > ### Level 2 ToM Expression
> > > > >
> > > > > $b\_{i,R}^2(a\_j^R) = p(a\_j^R|\\text{ToM}\_i^1(o\_i^{1:R}, \hat{a}\_{-i}^{1:R-1}, \\{b^0\_{i,R}(a_m^R)\\}\_{m \neq i}))$
> > > > >
> > > > > ### Step-by-Step Reasoning Process:
> > > > >
> > > > > 1. **Base Knowledge (Level 0)**
> > > > >     - i observes the current game state $o_i^{1:R}$
> > > > >     - i knows the history of moves $\hat{a}_{-i}^{1:R-1}$ including i's own past moves
> > > > >     - This forms i's base beliefs $b^0_{i,R}(a_m^R)$
> > > > > 2. **First Layer of Reasoning (Level 1)**
> > > > >     - i thinks: "Given my previous aggressive knight moves..."
> > > > >     - i forms beliefs about j's thinking: "j has seen my aggressive style"
> > > > >     - This is captured in $\\text{ToM}\_i^1$ *which uses history $\\hat{a}\_{-i}^{1:R-1}$*
> > > > > 3. **Second Layer of Reasoning (Level 2)**
> > > > >     - i thinks: "j probably thinks I will make another aggressive move"
> > > > >     - i reasons: "j is likely preparing a defensive response because they think I'll be aggressive"
> > > > >     - This nested thinking is captured in $b\_{i,R}^2(a\_j^R)$
> > > > > 4. **Prediction Formation**
> > > > >     - Based on this recursive reasoning, i predicts j's action $a\_j^R$
> > > > >     - Example: "j will likely position their bishop defensively because they think I'm planning another aggressive knight move"

---

> > > > > > ### Author Response · Authors · 2024-11-27
> > > > > > **Summary of rebuttal (including main concerns, our responses and improvements)**
> > > > > >
> > > > > > We sincerely thank the reviewer for their thorough and constructive feedback. Below is a comprehensive summary addressing both initial concerns (W1-W5), initial questions (Q1-P2) and follow-up questions (R1-P5, P1-P3).
> > > > > >
> > > > > > ## 1. Theory of Mind (ToM) Formulation (W1&2, R1&2, P3, Q1)
> > > > > >
> > > > > > > **Main Concerns:**
> > > > > > >
> > > > > > - Deviation from common higher-order ToM definition
> > > > > > - Unclear belief representation for different ToM inferences
> > > > > > - How does agent i access other agents' hidden mental states?
> > > > > >
> > > > > > > **Our Responses:**
> > > > > > >
> > > > > >
> > > > > > ### ToM Definition and Implementation
> > > > > >
> > > > > > - Clarified recursive ToM formulation for agent i at round R: $ToM\_i^k(o\_i^{1:R}, â\_{-i}^{1:R-1}, \\{b^{k-1}\_{i,R}(a_m^R)\\}\_{m≠i}) := b\_{i,R}^k$
> > > > > > - No direct access to others' mental states; beliefs derived from observations
> > > > > > - The reviewer seems misunderstand the “other’s beliefs” in high level ToM. Recursive reasoning does not rely on “other’s actual belief”.
> > > > > > - Demonstrated ToM reasoning through chess game example for agent i:
> > > > > >     - Level 0: Direct observations of game state and move history
> > > > > >     - Level 1: "j has seen my aggressive style" (this is agent i’s belief, not j’s)
> > > > > >     - Level 2: "i think j probably thinks I will make another aggressive move"
> > > > > >
> > > > > > *Manuscript Updates***:**
> > > > > >
> > > > > > - Clarify ToM formulation (Pages 4-5)
> > > > > > - Clarified belief structure and update mechanisms (Section 4.1)
> > > > > > - Added discussion for non-LLM agents computing belief alignment(Appendix A. Remarks)
> > > > > >
> > > > > > ## 2. Multi-Agent System Design (W3, W5, P1, Q2)
> > > > > >
> > > > > > > **Main Concerns:**
> > > > > > >
> > > > > > - Unclear agent configuration
> > > > > > - Questions about prompt dependency
> > > > > > - Concerns about experimental fairness
> > > > > > - Impact of specialized skill scores
> > > > > > - Limited applicability to partially observable environments
> > > > > >
> > > > > > > **Our Responses:**
> > > > > > >
> > > > > >
> > > > > > ### System Configuration
> > > > > >
> > > > > > Detailed settings across tasks (minimum coalition size is ⌈N/2⌉):
> > > > > >
> > > > > > 1. Programming Task:
> > > > > >     - 5 agents (1 PM + 4 Engineers)
> > > > > >     - Coalition size of 3
> > > > > > 2. Debate Task:
> > > > > >     - 6 agents (3 per side)
> > > > > >     - 2 active debaters per round in ALL conditions
> > > > > > 3. Logical Reasoning:
> > > > > >     - 3 agents (ToM levels 0,1,2)
> > > > > >     - Coalition size of 2
> > > > > >
> > > > > > ### Implementation Clarification
> > > > > >
> > > > > > - Algorithm is a "plug-and-play" mechanism independent of prompts
> > > > > > - Information asymmetry is not our current focus
> > > > > > - Same matching algorithm applied across different tasks
> > > > > > - Fair comparison maintained through consistent agent numbers
> > > > > >
> > > > > > *Manuscript Updates***:**
> > > > > >
> > > > > > - Added case study for specialized scores (Appendix C.1)
> > > > > >
> > > > > > ## 3. Evaluation and Metrics (W4, R4, P2)
> > > > > >
> > > > > > > **Main Concerns:**
> > > > > > >
> > > > > > - Questions about evaluation metrics
> > > > > > - Apparent conflict in cooperation trend results
> > > > > > - R4: Concerns about circular reasoning in evaluation metrics
> > > > > > - R2: Questions about performance improvements across rounds
> > > > > >
> > > > > > > **Our Responses:**
> > > > > > >
> > > > > >
> > > > > > ### Comprehensive Evaluation
> > > > > >
> > > > > > Added performance analysis across ToM configurations:
> > > > > >
> > > > > > 1. Initial Performance (Round 1):
> > > > > >     - Similar baseline across configurations
> > > > > >     - Example: HumanEval scores 0.87-0.93
> > > > > > 2. Without Matching (Round 5):
> > > > > >     - Performance degradation observed
> > > > > >     - PM(ToM=2) dropped from 0.90 to 0.85 on HumanEval
> > > > > > 3. With Matching (Round 5):
> > > > > >     - PM(ToM=2) + Eng(ToM=0) achieved best results:
> > > > > >         - HumanEval: 0.96 (+12.9%)
> > > > > >         - MBPP: 0.60 (+22.4%)
> > > > > >
> > > > > > ### Response to P2
> > > > > >
> > > > > > - Clarified our goal is not adding ToM but optimizing existing capabilities
> > > > > > - Matching mechanism helps ToM agents to achieve better cooperation outcomes.
> > > > > >
> > > > > > *Manuscript Updates***:**
> > > > > >
> > > > > > - Added comprehensive evaluation results in Appendix F.3

---

> > > > > ### Author Response · Authors · 2024-12-01
> > > > > **Close to the end of discussion December 2nd**
> > > > >
> > > > > We sincerely appreciate your time and effects during the review and rebuttal period.
> > > > >
> > > > > As we approach the end of the rebuttal period on December 2nd, we wanted to follow up our previous discussion. We hope our response has adequately addressed your concerns. If any points require further clarification, we would be grateful for your feedback. Thank you again for your consideration throughout this process.

---

### Official Review · Reviewer_BmBs · 2024-11-04

**Soundness:** 3
**Presentation:** 3
**Contribution:** 3
**Rating:** 6
**Confidence:** 3

**Summary:**

The authors present a method for using using Theory of Mind (ToM) and a coalition matching algorithm to allow LLM agents (using various LLM models) to cooperatively perform tasks in environments such as:

-Iterative Programming (HumanEval, MBPP)

-Debate (Two cooperative teams compete, with affirmative team taking on various forms of the model (no-ToM, ToM without matching, ToM with matchin), and negative team takes the baseline no-ToM)

-Logical and General Reasoning (Using AQUA-RAT and MMLU datasets)

The k-level ToM is set to take in an observation, the action of all agents at the previous timestep and the belief of the actions of all agents at the previous timestep, at the 0-level this is set to start with no belief. These are open-ended action spaces defined by natural language, and the observations, actions and beliefs are textual outputs. (The prompts of these are demonstrated in the appendix)

The Matching coalition algorithm takes a set of LLM agents and the possible matchings of these agents. It then assigns a preference order of these matchings. It aims to create stable matchings based on this preference order, such that agent i prefers agent j over all other pairings and agent j prefers agent i and neither agent has incentive to deviate from this pairing. A specific rule for this preference order is define based on the alignment, based on semantic similarity as calculated by the LLM, between beliefs of actions and the actual actions, and the agents are only matched if this  is above a certain threshold.

The results show that without matching lower ToM levels have higher cooperative trends, while with matching higher ToM levels have better cooperative trends. In all shown environments the ToM w. Matching  (their method) outperforms the baselines of no-ToM, or ToM w.o. matching.

**Strengths:**

The paper is easy to follow, with the appendix clearly aiding in understanding how the models function.

There is a clear logic as to why each component is added, this is shown through the experimentation and the results. Especially the need for adding coalition matching on top of the theory of mind.

There is a clear increase in Pass@1 the iterative programming environment with this model.

There is a clear increase in accuracy in the logic and reasoning problems compared to existing methods.

**Weaknesses:**

The calculation of the semantic similarity of beliefs and actions is left to the LLM, this does not lend itself to a general approach as the title alludes to. It is made clear throughout the paper that this is applied to LLMs however and I do not see this as a big weakness, but would like to see this made clear in the title if possible.

In the debate environment the baselines where both affirmative and negative lead to a bias of the affirmative winning 65.45% of the time, that is they are both using the same method. This is a cause for concern that this result may not be robust enough and might simply be taking advantage of this bias, is it possible to show the results the other way around? (With your model placed in the negative.)

Minor: The acronym FTM (Fraction of trust members/Frequency of team matching) is used multiple times making some sections difficult to understand.

**Questions:**

1)	For a non-LLM environment, how will the matching scores be calculated?
2)	In the debate environment the baselines where both affirmative and negative lead to a bias of the affirmative winning 65.45% of the time, that is they are both using the same method. This is a cause for concern that this result may not be robust enough and might simply be taking advantage of this bias, is it possible to show the results the other way around? (With your model placed in the negative.)

---

> ### Author Response · Authors · 2024-11-19
>
> We sincerely thank reviewer BmBs for highlighting our contribution of combining coalition matching with Theory of Mind.
>
> > W1. The calculation of the semantic similarity of beliefs and actions is left to the LLM, this does not lend itself to a general approach as the title alludes to. It is made clear throughout the paper that this is applied to LLMs however and I do not see this as a big weakness, but would like to see this made clear in the title if possible.
>
> R1. We appreciate the reviewer's comments about the semantic similarity calculation and its implications for generalizability.
>
> - For LLM-based agents, our approach **leverages one of the key advantages** of LLMs: their ability to handle **open-ended trajectories**. The self-evaluation method allows LLMs to assess alignment in complex, unstructured action spaces where traditional similarity metrics might fall short. This is particularly valuable in scenarios involving natural language interactions and creative problem-solving, where the action space cannot be easily enumerated.
>
> - For **non-LLM environments**, our matching mechanism could be adapted more straightforwardly due to the typically more **constrained and well-defined action spaces**. For example, in game environments, actions could be discrete choices or continuous control signals. In such cases, belief-action alignment could be computed using **traditional similarity metrics** like Euclidean distance, KL divergence, or task-specific reward functions. While this would require **more careful design of the action space** for each specific task, the computation of alignment scores would be more straightforward and computationally efficient.
>
> We will make this distinction clearer to better reflect the current focus of our work, and we propose modifying the title to explicitly mention LLM-based agents.
>
>
> > W2. In the debate environment the baselines where both affirmative and negative lead to a bias of the affirmative winning 65.45% of the time, that is they are both using the same method. This is a cause for concern that this result may not be robust enough and might simply be taking advantage of this bias, is it possible to show the results the other way around? (With your model placed in the negative.)
>
> R2. We thank the reviewer raise a valid concern about the affirmative side bias in our debate experiments. To address this, we conducted additional experiments with our model on the negative side:
>
> | Setting | Win Rate |
> | --- | --- |
> | No-ToM | 34.55% (65.45% Win Rate for the affirmative side) |
> | ToM w.o. Matching | 25.45% |
> | ToM w. Matching (Ours) | **36.36%** |
>
> **Remarks**: These results demonstrate that:
>
> - The coalition matching mechanism provides **robust benefits regardless of the debate side**.
> - The improvement over baseline is **consistent with our original findings**.
>
> > Minor: The acronym FTM (Fraction of trust members/Frequency of team matching) is used multiple times making some sections difficult to understand.
>
> We will revise the paper to maintain "FTM" only for "Fraction of Trusted Members".
>
> > Q1. For a non-LLM environment, how will the matching scores be calculated?
>
> Please refer to our response R1 to W1.
>
> > Q2. In the debate environment the baselines where both affirmative and negative lead to a bias of the affirmative winning 65.45% of the time, that is they are both using the same method. This is a cause for concern that this result may not be robust enough and might simply be taking advantage of this bias, is it possible to show the results the other way around? (With your model placed in the negative.)
>
> Please refer to our response R2 to W2.

---

> > ### Comment · Reviewer_BmBs · 2024-11-23
> >
> > Reply to R1:
> > I am happy with this explanation, and would be willing to defend a change to your title to explicitly mention LLMs to align with your content should this be accepted and the title change allowed.
> >
> > Reply to R2:
> > Changing the position of the agent to being on the negative and the results shown do match the remarks made, and I believe including this would likely lead to a better understanding of the contribution of the work.
> >
> > While this specific debate environment does appear to have a bias for the affirmative side winning, these results do address my concerns about the method leveraging these biases. The findings shown here are consistent with that of the affirmative side, that there is a decrease in win rate with only ToM, and an increase in win rate when both coalition matching and ToM are included.
> >
> > This shows that ToM is not enough on its own to improve the win rates. In fact, it appears to make it worse in both cases, affirmative or negative.
> > This highlights the contribution that both coalition matching + ToM are both needed, and that this addition does improve the win rates by a consistent amount over no-ToM in both cases.
> >
> > Minor: Maintaining that FTM is only used for Fraction of Trusted Members is a welcomed change.

---

> > > ### Author Response · Authors · 2024-11-23
> > >
> > > We sincerely thank you for your positive feedback on our manuscript, and greatly appreciate your careful consideration of our responses and the additional experimental results we provided.
> > > As suggested, we have explicitly mentioned LLMs in our title to better align with the paper's content. In our updated manuscript, the title is modified to "Cognitive Insights and Stable Coalition Matching for Fostering **LLM-based** Multi-Agent Cooperation."
> > >
> > > Given your positive assessment of how we've addressed your concerns, we respectfully ask if you would consider raising your score?
> > >
> > > Thank you again for your detailed review and for helping us improve the clarity and rigor of our work!

---

### Official Review · Reviewer_zFGd · 2024-11-05

**Soundness:** 3
**Presentation:** 3
**Contribution:** 3
**Rating:** 6
**Confidence:** 2

**Summary:**

This paper focuses on the problem of cooperation in multi-agent systems when the agents are LLM agents. In particular, this work focuses on how theory of mind interacts with cooperation and introduce a mechanism for designing diverse ToM groups amongst these agents that optimise the overall cooperative performance.

**Strengths:**

- The related work is clear and concise.
- The paper is well motivated
- The empirical results for HumanEval demonstrating the effectiveness of the matching mechanism to match agents to those that they are able to accurately predict beliefs about is promising. This is alongside promising improvements in terms of Pass@1 rates.
- The empirical results are similarly promising in terms of problem solving and general reasoning.

**Weaknesses:**

- Whilst the authors do mention that the coalition formation is generally an NP-hard problem, they do not offer any ideas about potential future possibilities that would help with the scalability of the framework
- I do not understand the prompt referenced in Appendix A and the corresponding LLM output. The belief model is rather vague, and when looking at the output of the alignment scores it seems a bit arbitrary - e.g. the belief model does not mention using an object oriented approach, but in the alignment score this seems to be highly valued? I am just slightly concerned that some of the alignment scores outputted by the LLMs are not particularly strong signals and ideally it would be measured using something more robust.
- Overall, my main concern is the potential scalability of the proposed framework, with firstly the coalition forming being difficult and secondly the requirement to generate beliefs over all other agents. Furthermore, whilst the empirical results are good and I am not downplaying them, I am not convinced the proposed settings are those that can really leverage ToM fully. However, this is not impacting my score.

**Questions:**

- For the insight that low ToM exhibits better cooperation compared to high ToM, I wonder how specific this is to the environment being looked at. For example, the multi-agent programming setting, at least to me, does not strike me as an environment that requires much ToM to successfully cooperate in, therefore low ToM being more successful may simply be due to the lower complexity of using it. Have the authors noticed this same trend in other environments?

---

> ### Author Response · Authors · 2024-11-19
>
> We thank reviewer zFGd's recognition of our paper's clear motivation, novelty and promising results.
>
> We address specific concerns below point by point:
>
> > W1. Whilst the authors do mention that the coalition formation is generally an **NP-hard problem**, they do not offer any ideas about potential future possibilities that would help with the scalability of the framework
>
>
>   We thank the reviewer for raising this important point about scalability. We propose to add a dedicated discussion of scalability solutions in our future work section (section 7).
>
> - For example, we can employ **preference list truncation** where agents only maintain preferences for **a bounded number of potential partners**, reducing the complexity from O(n²) to O(kn) where k is a fixed constant much smaller than n.
> - Additionally, for future work, we can implement a **hierarchical matching** approach where agents are first grouped into clusters based on their ToM levels and task requirements, and then matching is performed within these **smaller clusters**. This would reduce the search space.
>
> > W2. I do not understand the prompt referenced in Appendix A and the corresponding LLM output. The belief model is rather vague, and when looking at the output of the alignment scores it seems a bit arbitrary - e.g. the belief model does not mention using an object oriented approach, but in the alignment score this seems to be highly valued? I am just slightly concerned that some of the alignment scores outputted by the LLMs are not particularly strong signals and ideally it would be measured using something more robust.
> > Overall, my main concern is the potential scalability of the proposed framework, with firstly the coalition forming being difficult and secondly the requirement to generate beliefs over all other agents. Furthermore, whilst the empirical results are good and I am not downplaying them, I am not convinced the proposed settings are those that can really leverage ToM fully. However, this is not impacting my score.
>
>
> We acknowledge the reviewer's concern about the robustness of alignment scores and propose to clarify:
>
> - **Alignment Score Measurement:**
> For LLM-based agents, our approach leverages one of the key advantages of LLMs: their ability to handle **open-ended trajectories** and perform nuanced semantic comparisons. The self-evaluation method allows LLMs to assess alignment in complex, unstructured action spaces where traditional similarity metrics might fall short.
> - **Alternative Approaches:**
> For non-LLM environments with **well-defined action spaces**, traditional metrics (e.g., Euclidean distance) could be used instead. We acknowledge this as an important direction for future work and will discuss alternative measurement approaches in our revised paper.
> -  We want to emphasize that our core contribution is the **coalition formation** mechanism as a **plug-and-play** approach for improving multi-agent cooperation with ToM agents.
>
> > Q1. For the insight that low ToM exhibits better cooperation compared to high ToM, I wonder how specific this is to the environment being looked at. For example, the multi-agent programming setting, at least to me, does not strike me as an environment that requires much ToM to successfully cooperate in, therefore low ToM being more successful may simply be due to the lower complexity of using it. Have the authors noticed this same trend in other environments?
>
> Our evaluation utilized different settings:
>
> - **Programming Tasks**: Despite being relatively *structured*, we observed higher ToM agents exhibiting overthinking and reduced cooperation.
> - **Debate Setting** (Section 6.4): Even in this highly *social* environment, we found higher ToM agents sometimes overthinking and showing less cooperative behavior initially.
> - We agree this opens interesting directions for future research on how environmental complexity interacts with ToM levels.

---

> > ### Author Response · Authors · 2024-11-24
> >
> > We sincerely thank the reviewer's thoughtful comments and suggestions. As we approach the end of the discussion period, we wanted to respectfully follow up regarding your concerns.
> >
> > We have provided detailed responses to address these concerns, including explanation of future scalability approaches and the additional analysis of ToM effectiveness across different environments.
> >
> > Please let us know if you have any remaining concerns. Thanks again for your thoughtful and constructive feedback making our paper clearer.

---

### Meta-Review · Area_Chair_TAdr · 2024-12-20

**Metareview:**

This paper is about theory of mind with LLM agents, and whether it helps them to form and manage coalitions of other LLM agents. Unfortunately, issues coming mainly from the game theory perspective, emerged during the review process. Two of the reviewers had extensive back and forth conversations with the authors but decided to maintain their low scores in the end.

**Additional Comments On Reviewer Discussion:**

There was quite a bit of conversation on this paper with two of the reviewers being quite engaged, asking several follow up questions, which the authors replied to. However, both of these reviewers decided to maintain their low scores in the end.

---

### Decision · Program_Chairs · 2025-01-22

Reject